# Neural P$^3$M: A Long-Range Interaction Modeling Enhancer for Geometric GNNs

**Yusong Wang**[1]$^*$, **Chaoran Cheng**[2]$^*$, **Shaoning Li**[3]$^*$, **Yuxuan Ren**[4]
**Bin Shao**[5], **Ge Liu**[2], **Pheng-Ann Heng**[3], **Nanning Zheng**[1]$^\dagger$

[1] National Key Laboratory of Human-Machine Hybrid Augmented Intelligence,
National Engineering Research Center for Visual Information and Applications,
and Institute of Artificial Intelligence and Robotics, Xi'an Jiaotong University
[2] University of Illinois Urbana-Champaign
[3] Department of Computer Science and Engineering, The Chinese University of Hong Kong
[4] University of Science and Technology of China
[5] Microsoft Research AI4Science

`wangyusong2000@stu.xjtu.edu.cn`, `{chaoran7, geliu}@illinois.edu`
`{snli24, pheng}@cse.cuhk.edu.hk`, `binshao@microsoft.com`
`nnzheng@mail.xjtu.edu.cn`

## Abstract

Geometric graph neural networks (GNNs) have emerged as powerful tools for modeling molecular geometry. However, they encounter limitations in effectively capturing long-range interactions in large molecular systems due to the localization assumption of GNN. To address this challenge, we introduce **Neural P$^3$M**, a versatile enhancer of geometric GNNs to expand the scope of their capabilities by incorporating mesh points alongside atoms and reimaging traditional mathematical operations in a trainable manner. Neural P$^3$M exhibits flexibility across a wide range of molecular systems and demonstrates remarkable accuracy in predicting energies and forces, outperforming on benchmarks such as the MD22 dataset. It also achieves an average improvement of 22% on the OE62 dataset while integrating with various architectures. Codes are available at https://github.com/OnlyLoveKFC/Neural_P3M.

## 1 Introduction

Prevailing geometric graph neural networks (GNNs) have demonstrated remarkable capabilities in capturing the geometric information inherent within molecular graphs. Not only do they accelerate the computational efficiency compared to traditional Density Functional Theory (DFT) methods for molecules, but also hold the promise of achieving high-level accuracy in predicting crucial molecular properties such as energy and forces [2, 18, 23]. Despite their success in modeling small molecules, limitations still persist in extending these methods to larger molecular structures and systems governed by periodic boundary conditions (PBC). Current methods [16, 1] excel in approximating the *short-range* interactions, which encapsulate interactions among local atom groups within a defined distance cutoff, characterized by a rapid decay in real space. The primary obstacle lies in effectively capturing *long-range* interactions within these complex systems.

Several attempts have been undertaken to incorporate long-range physical interactions into geometric GNNs. Early studies [19, 21] combined physical equations, such as Coulomb's law, with models

---

$^*$Equal contribution.
$^\dagger$Corresponding author.

38th Conference on Neural Information Processing Systems (NeurIPS 2024).

tailored for short-range interactions. Conversely, recent advancements are steering towards the development of sophisticated models capable of learning long-range interactions directly from data. One such strategy is the *spatial-based* method, exemplified by LSRM [13]. It utilizes specific fragmentation algorithms like BRICS [5] to fragment molecules into discrete groups in real space. The long-range interactions are thereby captured in a hierarchical manner by facilitating message passing between the fragments and atoms. Another strategy is the *spectral-based* method [12, 24], which treats the long-range parts in the reciprocal space following the concepts of Ewald summation [4]. The long-range parts exhibit a rapid decay instead in the reciprocal space, which enables efficient evaluation with a frequency cutoff.

Following traditional computational chemistry, an intuitive direction would be to mesh up the Ewald summation, harnessing fast Fourier transformation (FFT) for acceleration. While this poses a non-trivial problem, a rich of established works represented by $\underline{P}$article–$\underline{P}$article $\underline{P}$article-$\underline{M}$esh ($P^3M$) [11] provide a solid foundation for such undertakings. In this work, inspired by the underlying unified concepts [6] behind these FFT-accelerated methods, we propose a novel perspective by integrating *atom* and *mesh* into neural networks. To be concrete, we reimage the traditional mathematical operations in mesh-based methods in a trainable manner, laying the foundation of our new framework, termed **Neural $P^3M$** (Fig. 1). Neural $P^3M$ is designed to be a versatile enhancer, compatible with a wide range of existing models. In contrast to LSRM, Neural $P^3M$ framework remains unconstrained to any fragmentation algorithm, and hence enhances its flexibility across diverse molecular systems. Different from Ewald MP, Neural $P^3M$ explicitly incorporates mesh representations, thereby offering discrete resolutions necessary for formulating long-range terms. Additionally, it incorporates the exchange of information between short-range and long-range terms at the atom and mesh scales. Moreover, our proposed framework exhibits theoretical efficiency surpassing that of Ewald MP due to the reduced computational complexity afforded by FFT.

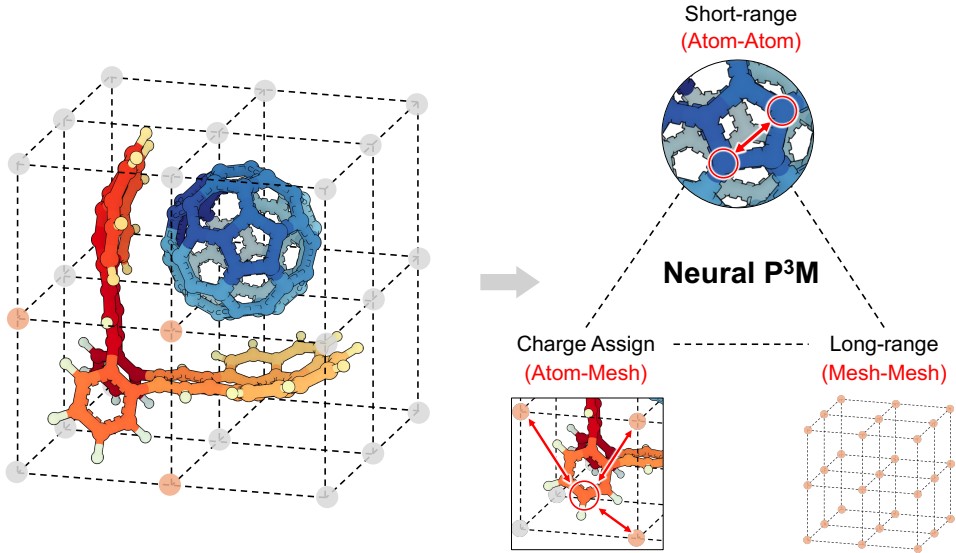

Figure 1: Illustration of Particle–Particle Particle-Mesh ($P^3M$) and its relationship with our Neural $P^3M$ framework. The **Atom2Atom** block corresponds to the short-range term. The **Atom2Mesh** and **Mesh2Atom** block are similar to the charge assignment and back-interpolation. The **Mesh2Mesh** block corresponds to the long-range term.

We evaluate our framework on several benchmarks by integrating a variety of geometric GNNs. Neural $P^3M$ achieves the state-of-the-art performance on the MD22 dataset [3] and Ag dataset [16] when combined with ViSNet [23]. It consistently demonstrates improvements in energy mean absolute errors (MAEs), achieving an average reduction of 22% on the OE62 dataset [20]. In summary, our contributions can be summarized as follows:

- **Framework.** We propose a novel framework **Neural P$^3$M** to capture *short-range* and *long-range* interactions at both *atom* and *mesh* scale.

- **Enhancement and Versatility.** Neural P$^3$M exhibits compatibility and significant improvements with short-range-centric methods on the Ag, MD22 and OE62 benchmarks.

- **Flexibility.** Neural P$^3$M is well-suited for diverse molecular systems without any constraints.

## 2  Preliminary

**Ewald Summation**    Ewald summation is a widely used technique in calculations of long-range interactions in periodic systems [7]. Specifically, consider the pair-wise electrostatic potential as $\psi(\mathbf{r}_{ij}) = 1/\|\mathbf{r}_{ij}\|_2$. The total electrostatic potential energy $E$ can be evaluated as the infinite summation over pairs under the periodic boundary condition (PBC) as

$$E = \frac{1}{2} \sum_{\mathbf{n}} \sum_{i=1}^{N} \sum_{j=1}^{N}{}' \iint \rho_i(\mathbf{r})\rho_j(\mathbf{r}')\psi(\|\mathbf{r} - \mathbf{r}' + \mathbf{n} \cdot \mathbf{c}\|_2)d^3\mathbf{r}d^3\mathbf{r}' = \frac{1}{2} \sum_{i=1}^{N} \int \rho_i(\mathbf{r})\,\phi_{[i]}(\mathbf{r})d^3\mathbf{r} \quad (1)$$

where $\rho_i(\mathbf{r})$ is charge density, $\mathbf{c}$ is the cell vector, and $N$ is the number of atoms in a cell. The $'$ summation is introduced to exclude the term $j = i$, if and only if $\mathbf{n} = 0$. $\phi_{[i]}(\mathbf{r})$ represents the potential generated by all particles excluding the particle $i$. A continuous partition function that delays rapidly with respect to the distance is used to separate the short-range and long-range terms. One standard approach is to partition the contributions based on the error function $\mathrm{erf}$:

$$\psi^{\mathrm{sr}}(\mathbf{r}) = \frac{1 - \mathrm{erf}(\beta\|\mathbf{r}\|_2)}{\|\mathbf{r}\|_2}, \psi^{\mathrm{lr}}(\mathbf{r}) = \frac{\mathrm{erf}(\beta\|\mathbf{r}\|_2)}{\|\mathbf{r}\|_2} \quad (2)$$

where $\beta$ is a fixed constant. We assume the charge density is described by the delta function as point charges, i.e. $\rho_i(\mathbf{r}) = q_i\delta(\mathbf{r} - \mathbf{r}_i)$. With the rapid delay of the partition function, it is safe to assume convergence by only considering the interaction pairs within a specific cutoff distance as

$$E^{\mathrm{sr}} = \frac{1}{2} \sum_{i=1}^{N} \int \rho_i(\mathbf{r})\,\phi^{\mathrm{sr}}_{[i]}(\mathbf{r})d^3\mathbf{r} = \frac{1}{2} \sum_{(i,j)\in\mathcal{E}} q_i q_j \psi^{\mathrm{sr}}(\mathbf{r}_{ij}) \quad (3)$$

where $\mathcal{E}$ is the set of atom pairs within the cutoff distance. By the Parseval's theorem, the corresponding long-range term can be expressed as the summation in the Fourier domain as

$$E^{\mathrm{lr}} = \frac{1}{2} \sum_{i=1}^{N} \int \rho_i(\mathbf{r})\,\phi^{\mathrm{lr}}(\mathbf{r})d^3\mathbf{r} = \frac{1}{2V} \sum_{\mathbf{m}\neq 0} \tilde{g}(\mathbf{m})\tilde{\gamma}(\mathbf{m})\|\tilde{\rho}(\mathbf{m})\|_2^2 \quad (4)$$

where $V$ is the volume of the unit cell and $\tilde{g}(\mathbf{m}) = 4\pi/\|\mathbf{m}\|_2^2$ are the Fourier transformed Green function of the Coulomb potential $1/\|\mathbf{r}\|_2$, and $\tilde{\gamma}(\mathbf{m}) = \exp(-\|\mathbf{m}\|_2^2/4\beta^2)$. The Fourier-transformed charge density $\tilde{\rho}(\mathbf{m})$ is defined as

$$\tilde{\rho}(\mathbf{m}) = \int_V \rho(\mathbf{r})\,e^{-i\mathbf{m}\cdot\mathbf{r}}d^3\mathbf{r} = \sum_{j=1}^{N} q_j e^{-i\mathbf{m}\cdot\mathbf{r}_j} \quad (5)$$

The frequency vector $\mathbf{m}$ can be truncated as the long-range term quickly converges in the Fourier domain. As the long-range term introduces the self-interaction energy, a correction term is also applied to the final potential energy as

$$E^{\mathrm{self}} = -\frac{1}{2} \sum_{i=1}^{N} \int \rho_i(\mathbf{r})\,\phi^{\mathrm{lr}}_i(\mathbf{r})d^3\mathbf{r} = -\frac{\beta}{\sqrt{\pi}} \sum_{i=1}^{N} q_i^2 \quad (6)$$

**Meshing up the Ewald Summation**    The traditional Ewald summation method has a computational complexity of $O(N^2)$, which becomes impractical for large-scale systems. A common approach to accelerate the process is to employ FFT. Currently, a variety of mesh-based implementations are available. While they differ in their implementations, they share a similar conceptual foundation [6].

Initially, point charges (particles) with their continuous coordinates, must be scattered onto grid-based charge densities (meshes). The charge densities on meshes are interpolated using *charge assignment function $W$* to ensure a finite support for summation:

$$\rho_M(\mathbf{r}_p) = \frac{1}{V_{\text{grid}}} \int_V W(\mathbf{r}_p - \mathbf{r}) \, \rho(\mathbf{r}) d^3\mathbf{r} = \frac{1}{V_{\text{grid}}} \sum_{i=1}^{N} q_i W(\mathbf{r}_p - \mathbf{r}_i) \tag{7}$$

where $V_{\text{grid}}$ is the volume of the discrete grid to ensure that $\rho_M$ is a density. Once we have discrete grid-based charge densities, we need to modify Eq.4 to accommodate discrete mesh points. According to the proof in Appendix B, Eq.4 can be rewritten as the convolution in the real space:

$$E^{\text{lr}} = \frac{1}{2} \sum_{i=1}^{N} q_i \phi^{\text{lr}}(\mathbf{r}_i) = \frac{1}{2} \sum_{i=1}^{N} q_i [g \star \gamma \star \rho](\mathbf{r}_i) = \frac{1}{2} \sum_{i=1}^{N} q_i [G \star \rho](\mathbf{r}_i) \tag{8}$$

where $G$ is referred to *influence function* following Hockney and Eastwood [11] and $\star$ is the convolution operation. The discrete approximation for $E^{\text{lr}}$ can be expressed in a corresponding manner as follows:

$$E^{\text{lr}} \approx \frac{1}{2} \sum_{\mathbf{r}_p \in \mathcal{V}} V_{\text{grid}} \rho_M(\mathbf{r}_p)[G \star \rho_M](\mathbf{r}_p) \tag{9}$$

where, $\mathcal{V}$ is the set of mesh points. By altering the standard influence function $G$ to accommodate different charge assignment functions, one can develop distinct algorithms. Subsequently, FFT is employed to accelerate the convolution process. Following the calculation of the energy, forces on particles can be determined by differentiation, either in the real space or Fourier space. Alternatively, forces can also be derived by differentiating on meshes and then applying a *back-interpolation* technique to assign forces to particles.

The adaptation of FFT to the Ewald summation has been quite enlightening. We will delve into a detailed examination of the correlation between our Neural P$^3$M and these mesh-based techniques in the subsequent section.

## 3 Method

We are interested in learning the energies and forces of 3D molecules, potentially under the assumption of the periodic boundary condition. Specifically, consider a 3D molecule represented as a point cloud $\mathcal{G} = \{\mathbf{x}_i^a, z_i\}_{i \in \mathcal{U}}$ with atom coordinates $\mathbf{x}^a$ and atom types $z$, we want to learn the molecule-level energy $\hat{E}(\mathcal{G})$ and atom-level forces $\hat{F}(\mathcal{G})$. Different from previous work [12] which utilizes the vanilla Ewald summation in the Fourier domain, our framework is mesh-based which provides discrete structural information and allows for information flow between long-range and short-range representations. Our fundamental concept is akin to these mesh-based methods mentioned in Section 2. We use short-range blocks on atoms to capture bonded terms and non-bonded short-range terms while applying long-range blocks on meshes to handle long-range terms. We enable the transfer of information between atoms and meshes via the representation assignment. A pseudocode for the Neural P$^3$M block is provided in Appendix D.1 to enhance understanding. We further elaborate on the Neural P$^3$M architecture as follows.

### 3.1 Mesh Construction

Firstly, we construct meshes on which long-range interactions can be captured. In periodic systems such as crystals, the cell is naturally delineated. For non-periodic systems, we adopt the approach used by prevalent quantum chemistry software, which involves padding the bounding box with a specified margin to define the cell. Detailed information about the construction of the cell can be found in Appendix C. The coordinates of mesh points $\mathbf{x}_{i,j,k}^m$ can be described as:

$$\mathbf{x}_{i,j,k}^m = \frac{n_i + 1/2}{N_x} \mathbf{c}_x + \frac{n_j + 1/2}{N_y} \mathbf{c}_y + \frac{n_k + 1/2}{N_z} \mathbf{c}_z \tag{10}$$

where $\mathbf{c} = [\mathbf{c}_x, \mathbf{c}_y, \mathbf{c}_z]^\top$ is the cell vector and $N_x, N_y, N_z$ is the number of discretizations along each dimension. For convenience, we can regard meshes as a point cloud with a single subscript for the index as $\{\mathbf{x}_i^m\}_{i \in \mathcal{V}}$.

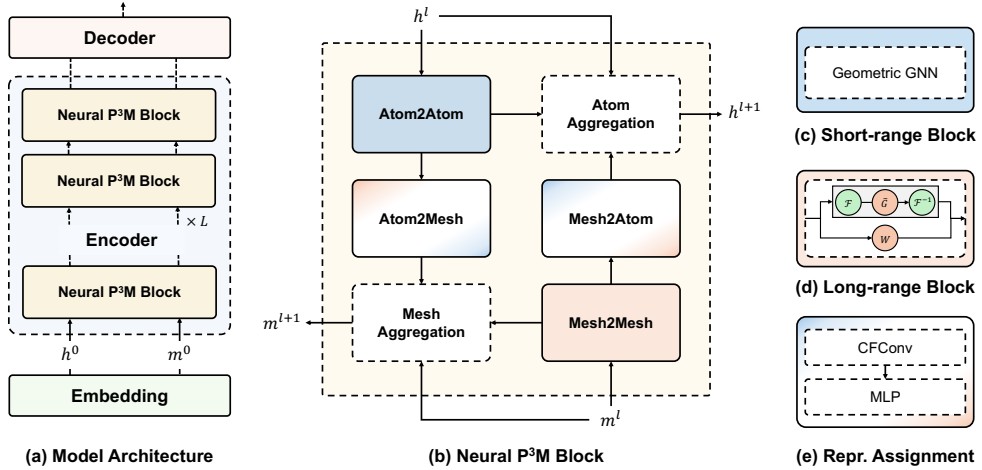

**(a) Model Architecture**  **(b) Neural P³M Block**  **(c) Short-range Block**  **(d) Long-range Block**  **(e) Repr. Assignment**

Figure 2: Overall framework architecture and details of each block. Geometric GNN models short-range interactions, Fourier neural operator (FNO) captures global long-range interactions, and continuous filter convolution (CFConv) exchanges information between two parts.

## 3.2 Embedding Block

Once coordinates of mesh points are established, we can proceed to construct a short-range atomic radius graph and a bipartite radius graph between atoms and meshes as follows:

$$\mathcal{E}^{\text{short}} = \{e_{ij} : \|\mathbf{x}_i^a - \mathbf{x}_j^a\|_2 \leq r^{\text{short}}, \forall i, j \in \mathcal{U}\}. \tag{11}$$

$$\mathcal{E}^{\text{assign}} = \{e_{ij} : \|\mathbf{x}_i^a - \mathbf{x}_j^m\|_2 \leq r^{\text{assign}}, \forall i \in \mathcal{U}, j \in \mathcal{V}\}. \tag{12}$$

where $\mathcal{U}$ is the atom set and $\mathcal{V}$ is the mesh set. Specifically, for periodic systems, the edges are also obtained by considering possible cross-boundary connections. The atom representation $h_i^0$ is initialized as:

$$h_i^0 = \text{Embed}(z_i) \tag{13}$$

The initial mesh representation, denoted as $m_i^0$, is obtained by averaging the representations of all neighboring atoms on the atom-mesh bipartite graph:

$$m_i^0 = \frac{1}{|\mathcal{M}(i)|} \sum_{j \in \mathcal{M}(i)} h_j^0 \tag{14}$$

where $\mathcal{M}(i)$ represents the set of neighboring nodes connected to mesh node $i$ within $\mathcal{E}^{\text{assign}}$. The edge features in both $\mathcal{E}^{\text{short}}$ and $\mathcal{E}^{\text{assign}}$ can be expanded via a set of radial basis functions (RBF):

$$f_{ij}^{\text{short}} = e^{\text{RBF}}(\|\mathbf{x}_i^a - \mathbf{x}_j^a\|_2), f_{ij}^{\text{assign}} = e^{\text{RBF}}(\|\mathbf{x}_i^m - \mathbf{x}_j^a\|_2) \tag{15}$$

## 3.3 Neural P³M Block

**Short-range Block** The short-range block (Fig.2(c)) updates the atomic representations using a graph neural network that is either SE(3)-equivariant or invariant. This process can be generally expressed as follows:

$$\tilde{h}^l = \text{GNN}(h^l, \mathcal{E}^{\text{short}}, f^{\text{short}}) \tag{16}$$

We noted that the usage of radius graphs inherits the localization assumptions in geometric GNNs and any node is only able to aggregation information from its direct geometric neighbors in one short-range block. Therefore, we naturally interpret it as capturing the short-range contribution to the energy and forces. As this part involves only atoms, we call such a module **Atom2Atom** which corresponds to the *particle-particle* part (short-range term) in the P³M.

**Long-range Block**  The long-range block (Fig.2(d)) updates mesh representations globally. Recalling Eq.9, the key aspect is to devise the influence function $G$ and utilize FFT along with the convolution theorem for efficient computation of the convolution. Within our framework, we parameterize $\tilde{G}$ directly in the Fourier domain, and the updated mesh representations can be described as:

$$\tilde{m}^l \leftarrow \sigma \left( W^{\text{long}} m^l + \left( \mathcal{F}^{-1}(\tilde{G} \cdot \mathcal{F}) \right)(m^l) \right) \tag{17}$$

where $\mathcal{F}, \mathcal{F}^{-1}$ are the Fourier transform and inverse Fourier transform on the discretized mesh, respectively. $\sigma$ is the activation function. $W^{\text{long}}$ and $\tilde{G}$ are the learnable weights that parameterize the operator in the real space and Fourier space. If we consider $m$ as a continuous function $v(m)$, our formulation coincides with the Fourier neural operators (FNOs) on the discretized continuous function. Similarly, as the long-range block only involves interactions within meshes, we call it **Mesh2Mesh**.

**Representation Assignment**  The representation assignment block (Fig.2(e)) allows for information flow between atom representations and mesh representations, effectively mixing short-range and long-range terms to obtain a more comprehensive descriptor of the molecule. By parameterizing the charge assignment function $W$ in Eq.7 and substituting the charge density with the atom representation $\tilde{h}_j^l$, we can derive the continuous filter convolution (CFconv) proposed in SchNet [17]. To elaborate further, we get additional mesh representations as:

$$(m \leftarrow a)_i^l = \text{MLP} \left( \sum_{j \in \mathcal{M}(i)} \tilde{h}_j^l \cdot W_{m \leftarrow a}^l f_{ij}^{\text{assign}} \right) \tag{18}$$

This **Atom2Mesh** module can be regarded as the information flow from the short-range part to the long-range part. Similarly, the **Mesh2Atom** module takes the same input and geometric graph but outputs additional atom representations $(a \leftarrow m)^l$, which could be viewed as the back-interpolation operation. It allows for the information flow in the inverse direction, from the long-range part to the short-range part. The long-range Mesh2Mesh module together with the Atom2Mesh and Mesh2Atom modules corresponds to the *particle-mesh* part (long-range term) in the P$^3$M.

Ultimately, as shown in Fig.2(b), we merge the information updated by each part itself with the normalized information received from the other part, and we also incorporate a residual connection to obtain the final output as:

$$h^{l+1} = h^l + \tilde{h}^l + \text{LN}((a \leftarrow m)^l) \tag{19}$$

$$m^{l+1} = m^l + \tilde{m}^l + \text{LN}((m \leftarrow a)^l) \tag{20}$$

### 3.4 Decoder Block

As we are interested in the prediction of molecule-level energies and atom-level forces, an additional decoder is applied to the final atom representations $h^L$ and mesh representations $m^L$ to get the atom-wise energies $h^{\text{out}}$ and mesh-wise energies $m^{\text{out}}$. We follow previous work to assume the additive property of energy to sum all atom-wise energies as the short part of the molecule energy $\hat{E}^{\text{short}}$.

$$\hat{E}^{\text{short}} = \sum_{j \in \mathcal{U}} h_j^{\text{out}} = \sum_{j \in \mathcal{U}} \text{MLP}(\text{LN}(h_j^L)) \tag{21}$$

We also sum all mesh-wise energies as the long part of the molecule energy $\hat{E}^{\text{long}}$.

$$\hat{E}^{\text{long}} = \sum_{j \in \mathcal{V}} m_j^{\text{out}} = \sum_{j \in \mathcal{V}} \text{MLP}(\text{LN}(m_j^L)) \tag{22}$$

The final potential energy is calculated as: $\hat{E} = \hat{E}^{\text{long}} + \hat{E}^{\text{short}}$. Furthermore, although direct prediction of forces is possible, we instead use the negative gradient of the energy as the prediction of forces: $\hat{F} = -\nabla_{\mathbf{x}} \hat{E}$. The final training objective is a weighted loss between energy and force:

$$\mathcal{L} = \lambda_E |E - \hat{E}|^2 + \frac{\lambda_F}{3N} \sum_{i=1}^{N} \left\| F_i + \nabla_{\mathbf{x}_i} \hat{E} \right\|^2 \tag{23}$$

## 4 Experiment

### 4.1 Experimental Setup

In this section, we conduct comprehensive validations of our Neural $P^3M$ framework using diverse datasets and configurations. First, we intuitively demonstrate the necessity of incorporating long-range interactions through a toy dataset Ag used in Allegro [16]. Subsequently, we integrate various geometric GNNs [17, 9, 18, 8, 23] with our Neural $P^3M$ framework on two prevalent datasets OE62 [20] and MD22 [3] to demonstrate versatility and effectiveness. All results are evaluated using mean absolute error (MAE) on test sets, and the baseline results are sourced directly from the corresponding papers. Unless stated otherwise, almost all hyperparameters align with the baseline GNNs. For a more comprehensive overview of hyperparameter settings and implementation details, please refer to the Appendix D and F.

### 4.2 Toy Dataset: Ag

The Ag dataset comprises 1,159 structures sampled from a 1,111K AIMD simulation [16]. These structures were generated from a bulk face-centered-cubic lattice with a vacancy, encompassing 71 atoms subject to periodic boundary conditions. For consistency with Allegro, we randomly split them into 950 structures for training, 50 structures for validation and the remaining structures for testing. As shown in Fig. 3, compared to the strictly local Allegro model, ViSNet, which has only one layer, offers slightly improved force prediction, yet the energy prediction significantly deteriorates. This may be caused by the fact that the model can only perform message passing once, with a lack of long-range interactions. Long-range interactions can be complemented in theory by raising the cutoff from 4.0 Å to 12.0 Å, but this does not work in practice, because it could potentially

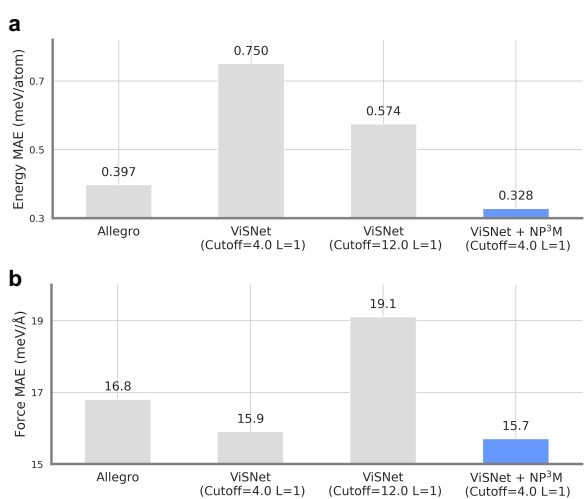

Figure 3: Mean absolute errors (MAEs) for energy and force predictions on Ag dataset are compared among Allegro, ViSNet, and our proposed framework.

lead to information over squashing problems, as mentioned in LSRM [13]. When ViSNet with a single layer is integrated into our framework, long-range interactions can be effectively captured, significantly improving the accuracy of energy and force predictions compared to the vanilla ViSNet and Allegro. This toy experiment intuitively demonstrates the critical need to incorporate long-range interactions and emphasizes the significance of a well-crafted methodology in incorporating them.

### 4.3 MD22

The MD22 dataset [3] consists of MD trajectory datasets, which present challenges due to their larger system sizes, ranging from 42 to 370 atoms. The number of structures in each molecule dataset ranges from 5,032 to 85,109. We calculate the diameter of each molecule, defined as the average of the maximum distance between any two atoms within a molecule. The smallest diameter observed is approximately 10.75 Å, while the largest molecule measures about 32.39 Å. We train a separate model for each molecule and randomly split the dataset according to sGDML [3].

Table 1 demonstrates the results of the ViSNet model incorporating with our Neural $P^3M$ framework (ViSNet-NP$^3$M for short) on MD22. ViSNet-NP$^3$M achieves the state-of-the-art (SoTA) performance on both energy and force predictions across the four largest molecules and also achieves the lowest mean absolute error (MAE) for energy or force predictions in the remaining three smaller molecules.

Table 1: Mean absolute errors (MAE) of energy (kcal/mol) and forces (kcal/mol/Å) for seven large molecules on MD22 compared with state-of-the-art models. The best one in each category is highlighted in **bold**.

| Molecule | Diameter (Å) | | sGDML | SO3KRATES | Allegro | Equiformer | MACE | ViSNet | | | |
| --- | --- | --- | --- | --- | --- | --- | --- | --- | --- | --- | --- |
| | | | | | | | | Baseline | Ewald | LSRM | Neural P³M |
| Ac-Ala3-NHMe | 10.75 | energy | 0.3902 | 0.337 | 0.1019 | 0.0828 | **0.0620** | 0.0796 | 0.0775 | 0.0654 | 0.0719 |
| | | forces | 0.7968 | 0.244 | 0.1068 | 0.0804 | 0.0876 | 0.0972 | 0.0814 | 0.0902 | **0.0788** |
| DHA | 14.58 | energy | 1.3117 | 0.379 | 0.1153 | 0.1317 | 0.1526 | 0.0932 | 0.0873 | 0.0781 | **0.0712** |
| | | forces | 0.7474 | 0.242 | 0.0732 | **0.0506** | 0.0646 | 0.0668 | 0.0664 | 0.0598 | 0.0679 |
| Stachyose | 13.87 | energy | 4.0497 | 0.442 | 0.2485 | 0.1404 | 0.1244 | 0.1283 | 0.1089 | 0.1055 | **0.0856** |
| | | forces | 0.6744 | 0.435 | 0.0971 | **0.0635** | 0.0876 | 0.0869 | 0.0976 | 0.0767 | 0.0940 |
| AT-AT | 17.63 | energy | 0.7235 | 0.178 | 0.1428 | 0.1309 | 0.1093 | 0.1688 | 0.1487 | 0.0772 | **0.0714** |
| | | forces | 0.6911 | 0.216 | 0.0952 | 0.0960 | 0.0992 | 0.1070 | 0.0885 | 0.0781 | **0.0740** |
| AT-AT-CG-CG | 21.29 | energy | 1.3885 | 0.345 | 0.3933 | 0.1510 | 0.1578 | 0.1995 | 0.1571 | 0.1135 | **0.1124** |
| | | forces | 0.7028 | 0.332 | 0.1280 | 0.1252 | 0.1153 | 0.1563 | 0.1115 | 0.1063 | **0.0993** |
| Buckyball catcher | 15.89 | energy | 1.1962 | 0.381 | 0.5258 | 0.3978 | 0.4812 | 0.4421 | 0.3575 | 0.4220 | **0.3543** |
| | | forces | 0.6820 | 0.237 | 0.0887 | 0.1114 | 0.0853 | 0.1335 | 0.0989 | 0.1026 | **0.0846** |
| Double-walled nanotube | 32.39 | energy | 4.0122 | 0.993 | 2.2097 | 1.1945 | 1.6553 | 1.0339 | 0.7909 | 1.8230 | **0.7751** |
| | | forces | 0.5231 | 0.727 | 0.3428 | 0.2747 | 0.2767 | 0.3959 | 0.2875 | 0.3391 | **0.2561** |

When compared to the vanilla ViSNet, ViSNet-NP³M showed an average improvement of 34.6% and 21.2% in energy and force prediction, respectively. Notably, our framework exhibits a more substantial improvement when compared to ViSNet-LSRM and ViSNet-Ewald, both of which utilize ViSNet as the short-range model. As shown in Appendix Table 5, another state-of-the-art model, Equiformer, when integrated with our Neural P³M framework, also demonstrates significant enhancements to the short-range model itself. These impressive results highlight our framework's ability to effectively improve the learning of potential long-range interactions in large molecules.

It's worth noting that for the two supramolecules that cannot be fragmented by LSRM, our Neural P³M achieves a significant performance improvement in energy prediction, with a 57.48% increase for the double-walled nanotube and a 16.07% increase for the buckyball catcher. This suggests that our Neural P³M is a general solution for various molecules, which is not limited by traditional fragmentation methods like BRICS.

## 4.4 OE62

We further take our analysis by incorporating four prevailing geometric GNNs including SchNet [17], PaiNN [18], DimeNet++ [9], and GemNet-T [8] on the OE62 dataset [20] to confirm the framework's versatility. The OE62 dataset consists of about 62,000 large organic molecules, each with the energy calculated by Density Functional Theory (DFT) . The structures within the OE62 dataset are non-periodic yet can span large spatial dimensions, exceeding 20 Å. The dataset is strictly split into train, validation, and test set according to Ewald MP [12]. The same dataset preprocessing process as Ewald MP is also applied.

The numerical results presented in Table 2 and Appendix Table 6 indicates that the Neural P³M framework, which combines four models, delivers more performance gains than Ewald MP and LSRM when using the same hyperparameters. Additionally, our framework exhibits a faster computation time than Ewald MP, likely due to the efficiency of FFT implementation by Pytorch. An unexpected observation is the speed performance of DimeNet++. Given that DimeNet++ does not inherently facilitate message passing between atom embeddings, Ewald MP compensates by integrating long-range interactions in each output block. In contrast, our approach exchanges short-range and long-range representations in each layer, which might account for our marginally slower speeds compared to Ewald MP. We also provide detailed profiling results for the number of model parameters, GPU memory usage, and other relevant metrics in Appendix G. For more details on the implementation on the four models, please refer to the Appendix D.

## 4.5 Ablation Study

### 4.5.1 Architecture

We first investigate the impact of the **Atom2Mesh** and **Mesh2Atom** modules. We remove the **Atom2Mesh** module from the original model to avoid the information flow from short-range blocks

Table 2: Energy MAEs and computation times per input structure for the OE62 dataset compared with Ewald MP and other baseline methods. The data was sourced directly from [12].

| Model | Variant | OE62-val | | OE62-test | | Forward Pass | | Forward & Backward Pass | |
|---|---|---|---|---|---|---|---|---|---|
| | | MAE meV ↓ | Rel. % ↑ | MAE meV ↓ | Rel. % ↑ | Runtime ms/struct. ↓ | Rel. % ↓ | Runtime ms/struct. ↓ | Rel. % ↓ |
| SchNet | Baseline | 133.5 | - | 131.3 | - | **0.13** | - | **0.28** | - |
| | Embeddings | 144.7 | -8.4 | 136.7 | -4.1 | 0.14 | 15.2 | 0.33 | 17.8 |
| | Cutoff | 257.4 | -92.8 | 254.8 | -94.1 | 0.14 | 13.6 | 0.31 | 11.6 |
| | SchNet-LR | 86.6 | 35.1 | 89.2 | 32.1 | 0.32 | 156.0 | 0.75 | 171.7 |
| | Ewald | 79.2 | 40.7 | 81.1 | 38.2 | 0.70 | 461.6 | 1.03 | 271.4 |
| | Neural P$^3$M | **70.2** | **47.4** | **69.1** | **47.4** | 0.37 | 184.6 | 0.57 | 103.6 |
| PaiNN | Baseline | 61.4 | - | 63.3 | - | **1.52** | - | **3.16** | - |
| | Embeddings | 63.5 | -3.4 | 63.1 | -0.2 | 1.54 | 1.4 | 3.28 | 3.8 |
| | Cutoff | 65.1 | -6.0 | 64.4 | -2.2 | 1.84 | 20.9 | 3.91 | 23.6 |
| | SchNet-LR | 58.3 | 5.1 | 58.2 | 7.7 | 1.84 | 20.7 | 4.21 | 33.1 |
| | Ewald | 57.9 | 5.7 | 59.7 | 5.7 | 2.29 | 50.5 | 4.57 | 44.4 |
| | Neural P$^3$M | **54.1** | **11.9** | **52.9** | **16.4** | 2.17 | 42.8 | 4.19 | 32.6 |
| DimeNet++ | Baseline | 51.2 | - | 53.8 | - | **1.99** | - | **4.26** | - |
| | Embeddings | 50.4 | 1.6 | 53.4 | 0.7 | 2.25 | 12.9 | 4.93 | 15.8 |
| | Cutoff | 48.3 | 5.7 | 48.1 | 10.6 | 2.68 | 34.7 | 6.10 | 43.4 |
| | SchNet-LR | 51.4 | -0.5 | 54.4 | -1.1 | 2.37 | 19.0 | 4.73 | 11.2 |
| | Ewald | 46.5 | 9.2 | 48.1 | 10.6 | 2.70 | 35.5 | 5.93 | 39.5 |
| | Neural P$^3$M | **40.9** | **20.1** | **41.5** | **22.9** | 3.11 | 56.3 | 5.62 | 31.9 |
| GemNet-T | Baseline | 51.5 | - | 53.1 | - | **3.07** | - | **6.96** | - |
| | Embeddings | 52.7 | -2.3 | 53.9 | -1.5 | 3.11 | 1.5 | 6.98 | 0.4 |
| | Cutoff | 47.8 | 7.2 | 47.7 | 10.2 | 4.02 | 31.2 | 8.88 | 27.7 |
| | SchNet-LR | 51.2 | 0.6 | 52.8 | 0.5 | 3.32 | 8.3 | 7.73 | 11.1 |
| | Ewald | 47.4 | 8.0 | 47.5 | 10.5 | 4.05 | 32.0 | 8.86 | 27.4 |
| | Neural P$^3$M | **47.2** | **8.3** | **47.4** | **10.7** | 3.93 | 28.0 | 7.71 | 10.8 |

to long-range blocks and vice versa. Table 3 demonstrates that both modules contribute synergistically to the model's overall performance. The results illustrate the necessity of enabling information exchange between the long-range and short-range blocks.

### 4.5.2 Hyperparameters

Compared to the vanilla model, our framework introduces only two new hyperparameters: the assignment cutoff distance between mesh points and atoms, denoted as $r^{\text{assign}}$, and the number of mesh points in each dimension, represented as $N_x, N_y, N_z$.

We find that the selection of the number of mesh points is crucial for the model's final performance. As illustrated in Appendix Fig. 4(b), the mean absolute error (MAE) in energy increases with the number of mesh points, while the forward computation time also extends. This decline in performance may be attributed to instances where each atom is assigned to multiple mesh points simultaneously. As such occurrences become more frequent, the model may struggle to effectively learn the appropriate assignment rules. In practice, we typically set the cutoff distance to either 4.0 or 5.0 Å, ensuring that the product of the number of mesh points and the cutoff distance is approximately equivalent to the cell size in each dimension.

Table 3: Energy MAE of SchNet-NP$^3$M variants on the OE62 test dataset. The best one is highlighted in **bold**.

| Architecture Variants | Energy MAE |
|---|---|
| Original | **69.10** |
| Without **Mesh2Atom** | 76.14 |
| Without **Atom2Mesh** | 74.48 |
| Without Both | 72.07 |

Additionally, we provide further ablation studies on the impact of the assignment cutoff distance (without the k-NN graph) to examine the effects of multiple assignments. As shown in Appendix Fig. 4(c), all experiments exhibit a slight decrease in performance due to multiple assignments. However, an appropriately chosen cutoff (4 or 5 Å) still yields relatively optimal results. Notably, the results do not worsen further as the assignment cutoff increases. We hypothesize that this may be because a larger assignment cutoff creates a broader neighborhood environment, facilitating the

learning of the assignment function with a fixed number of meshes, thereby alleviating the challenges associated with multiple assignments.

# 5 Related Work

**Geometric Graph Neural Networks**   Geometric graph neural networks preserve equivariance toward the rigid transformation in space, which can be categorized according to their emphasis on specific types of structural features and their respective methods of integration. SchNet [17] stands out as the pioneering approach to applying continuous filter convolution on molecular distances. Subsequently, DimeNet++ [9] and GemNet [8] explicitly incorporate angles and dihedrals using Fourier-Bessel functions. To address the computational complexity associated with angles extractions, PaiNN [18] and ViSNet [23] adopt the density trick and reduce the complexity to linear time. Additionally, many works are based on high-order geometric tensors [2, 1, 16, 22], which ensure rigorous theoretical guarantees of equivariance through the use of Clebsch-Gordan product. Despite these advancements, all these existing methods are constrained to the local atomic environment, and are unable to approximate the long-range interactions. Hence, there is an urgent need for a comprehensive framework to address this challenge.

**Long-range Interaction Modeling**   Incorporating long-range interactions into a short-range model is challenging. Early studies attempted to compensate these long-range effects by integrating physical equations with either hand-crafted terms [19] or predicted charges [21]. While, recent works have shifted towards creating carefully designed models that can directly learn long-range interactions from data. The LSRM framework [13], for instance, captures long-range interactions in real space by using specific algorithms to fragment molecules into discrete groups and models their interactions hierarchically. Other methods [12, 24, 15] handle long-range components in reciprocal space, employing concepts like Ewald summation [4]. Our approach differs from these works by introducing the discretized meshes and facilitating the exchange of information between long-range and short-range components.

# 6 Conclusion

In this paper, we introduce a novel framework, termed Neural $P^3M$, designed to enhance the long-range interaction modeling for various geometric GNNs. In addition, Neural $P^3M$ stands out by not being confined to any specific fragmentation approach, making it adaptable to various molecular systems. Neural $P^3M$ achieves significant performance improvement on prevalent benchmarks by capturing short-range and long-range interactions at both atom and mesh scales, and enabling the exchange of information between them.

**Limitation and Societal Impacts:** The limitation of our study is that it does not thoroughly investigate the impact of the number of meshes, nor does it explore potentially more effective methods for modeling long-range interactions beyond FFT. Nonetheless, our paper offers the community a fresh perspective on molecular geometry modeling. Our proposed Neural $P^3M$ framework is an extensive of existing geometric GNNs for energy and force prediction of molecules. The prediction of molecular energies and forces has diverse applications in downstream tasks including molecular dynamics simulation and molecular property prediction. As our framework better captures the long-range interaction within the molecule, it can potentially accelerate the pharmaceutical discovery and understanding of diverse molecules that have positive impacts on treating diseases. On the other hand, we are also aware of the potential negative impact if the model is misused, as our understanding of different molecules is still very limited. We will work closely with both the machine learning and the science community to ensure the proper usage of our model for the good of society.

# 7 Acknowledgments and Disclosure of Funding

We thank the reviewers for their valuable comments. This work was supported by NSFC under grant No. 62088102.

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

# Supplemental Material

## A Notations

Table 4: Glossary of notations

| Notation | Description |
|---|---|
| $i; j$ | The index of atoms or meshes |
| $l$ | The index of blocks |
| $\mathbf{r}, \mathbf{x^a}$ | The coordinates of particles (atoms) |
| $\mathbf{r}_p, \mathbf{x^m}$ | The coordinates of the meshes |
| $\mathbf{r}_{ij}$ | The displacement vector between the particle $i$ and $j$ |
| $\mathbf{c}$ | The cell vectors |
| $\mathbf{m}$ | The frequency vectors |
| erf | The error function |
| $\rho(\cdot)$ | The charge density of the particle |
| $\delta(\cdot)$ | The delta function |
| $q$ | The point charges |
| $\rho_M(\cdot)$ | The charge density of the mesh point |
| $\psi(\cdot)$ | The pair-wise electrostatic potential |
| $\phi(\cdot)$ | The potential generated by all particles |
| $\phi_{[i]}(\cdot)$ | The potential generated by all particles excluding the particle $i$ |
| $\phi_i(\cdot)$ | The potential generated by the particle $i$. |
| $\tilde{g}, \tilde{\gamma}, \tilde{\rho}$ | The Fourier transformed function $g, \gamma, \rho$ |
| $\star$ | The convolution operation |
| $N$ | The number of particles in a unit cell |
| $W$ | The charge assignment function |
| $G$ | The influence function |
| $V_{\text{grid}}, V$ | The volume of the discrete grid and cell. |
| $E, \hat{E}$ | The ground truth and prediction of potential energy |
| $F, \hat{F}$ | The ground truth and prediction of atomic forces |
| $z$ | The atom types |
| $\mathcal{U}, \mathcal{V}$ | The set of atoms and meshes |
| $N_x, N_y, N_z$ | The number of discretizations along each dimension $x, y, z$ |
| $r^{\text{short}}, r^{\text{assign}}$ | The cutoff distance of radius graphs |
| $\mathcal{E}^{\text{short}}, \mathcal{E}^{\text{assign}}$ | The edge set of radius graphs |
| $\mathcal{N}(i), \mathcal{M}(i)$ | The neighboring nodes of the target atom (mesh) node. |
| $h$ | The atom representations |
| $m$ | The mesh representations |
| $f$ | The edge representations |
| $\mathcal{F}, \mathcal{F}^{-1}$ | The Fourier transformer and inverse Fourier transform on the discretized mesh (FFT and IFFT) |
| $\| \cdot \|_2$ | The 2-norm of a vector |
| $\sigma(\cdot)$ | The activation function (SiLU) |
| $\text{GNN}(\cdot)$ | The short-range graph neural network (learnable) |
| $W^{\text{long}}$ | The weights in long-range block in real space (learnable) |
| $\tilde{G}$ | The weights in long-range block in Fourier space (learnable) |
| $W_{m \leftarrow a}, W_{a \leftarrow m}$ | The weights representation assignment (learnable) |
| $\text{MLP}(\cdot)$ | The multi layer perception (learnable) |
| $\text{LN}(\cdot)$ | The layer normalization |
| $\lambda_E, \lambda_F$ | The weights in the loss between energy and forces |

## B   Detailed Derivation of Eq.8

Let's start our derivation by replacing the the square of the $\tilde{\rho}(\mathbf{m})$'s modulus as the product of itself with its conjugate:

$$E^{\text{lr}} = \frac{1}{2V} \sum_{\mathbf{m} \neq 0} \tilde{g}(\mathbf{m})\tilde{\gamma}(\mathbf{m})\|\tilde{\rho}(\mathbf{m})\|_2^2 \tag{24}$$

$$= \frac{1}{2V} \sum_{\mathbf{m} \neq 0} \tilde{g}(\mathbf{m})\tilde{\gamma}(\mathbf{m})\tilde{\rho}(\mathbf{m})\tilde{\rho}^*(\mathbf{m}) \tag{25}$$

$$= \frac{1}{2V} \sum_{\mathbf{m} \neq 0} \tilde{g}(\mathbf{m})\tilde{\gamma}(\mathbf{m})\tilde{\rho}(\mathbf{m}) \sum_{j=1}^{N} q_j e^{i\mathbf{m}\cdot\mathbf{r}_j} \tag{26}$$

We can then confidently interchange the summation symbols and put the normalization factor $\frac{1}{V}$ within the summations as:

$$E^{\text{lr}} = \frac{1}{2V} \sum_{j=1}^{N} q_j \sum_{\mathbf{m} \neq 0} \tilde{g}(\mathbf{m})\tilde{\gamma}(\mathbf{m})\tilde{\rho}(\mathbf{m})e^{i\mathbf{m}\cdot\mathbf{r}_j} \tag{27}$$

$$= \frac{1}{2} \left( \sum_{j=1}^{N} q_j \left( \frac{1}{V} \sum_{\mathbf{m} \neq 0} \tilde{g}(\mathbf{m})\tilde{\gamma}(\mathbf{m})\tilde{\rho}(\mathbf{m})e^{i\mathbf{m}\cdot\mathbf{r}_j} \right) \right) \tag{28}$$

Using convolution theory, which states that the convolution of two functions is the pointwise product of their Fourier transforms, it becomes clear that the expression in parentheses represents the inverse Fourier transform. Consequently, we can rewrite the expression as follows:

$$E^{\text{lr}} = \frac{1}{2} \sum_{j=1}^{N} q_j [g \star \gamma \star \rho](\mathbf{r}_j) = \frac{1}{2} \sum_{i=1}^{N} q_j [G \star \rho](\mathbf{r}_j) \tag{29}$$

We refer $g \star \gamma$ as the smeared Coulomb Green function $G$ (influence function), and altering it when assigning charges with different charge assignment function $W$.

# C  Detailed Implementation of Cell Construction

Cell construction is trivial for periodic systems like crystals, as a canonical cell can always be assigned. We now describe the cell construction for non-periodic systems. Given a set of atom coordinates $\{\mathbf{x}_i\}_{i=1}^n$, we first derive a canonical coordinate frame $U$ as the eigenvectors of the covariance matrix:

$$U\Lambda U^\top = (X - \mu)^\top (X - \mu) \tag{30}$$

where $\Lambda$ is the diagonal matrix of the eigenvalues of the covariance matrix, $X \in \mathbb{R}^{n \times 3}$ is the coordinate matrix, and $\mu = \sum_{i=1}^n \mathbf{x}_i / n$. For any rotation matrix $R$ and $X' = XR$, it is easy to see that $U' = RU$ is a new eigenvector matrix for the new covariance matrix. Therefore, we use the canonical coordinates as $\tilde{X} = (X - \mu)U^\top$ which is invariant under global translation and rotation. After the transformation, the principle components of the coordinates now align with the coordinate frame. We can define the cell vectors to follow the directions of the coordinate with the cell length defined by the maximum coordinate span with additional padding $d$ on both sides:

$$\mathbf{c}_x = \left( \max_{1 \leq i \leq n} \tilde{x}_i - \min_{1 \leq i \leq n} \tilde{x}_i + 2d \right) \mathbf{e}_x$$

$$\mathbf{c}_y = \left( \max_{1 \leq i \leq n} \tilde{y}_i - \min_{1 \leq i \leq n} \tilde{y}_i + 2d \right) \mathbf{e}_y \tag{31}$$

$$\mathbf{c}_z = \left( \max_{1 \leq i \leq n} \tilde{z}_i - \min_{1 \leq i \leq n} \tilde{z}_i + 2d \right) \mathbf{e}_z$$

where $\tilde{x}, \tilde{y}, \tilde{z}$ are coordinate components of the transformed molecules. In practice, we used a $d = 0.5$Å. The mesh coordinates are obtained via Eq.10 and the final atom coordinates are obtained by moving the molecule inside the cell as:

$$Y = \tilde{X} - \left( \min_{1 \leq i \leq n} \tilde{x}_i - d, \min_{1 \leq i \leq n} \tilde{y}_i - d, \min_{1 \leq i \leq n} \tilde{z}_i - d \right) U \tag{32}$$

There are rare cases when the molecule exhibits high symmetry. However, as we only consider different atom types and treat the same type of atoms as indistinguishable, the final molecule and mesh are also indistinguishable and unique in this sense.

# D   Detailed Implementation for Integrating Various GNNs into Neural P$^3$M

In this section, we first provide the pseudocode for the Neural P$^3$M block to facilitate understanding of our framework, followed by a detailed explanation of the implementation. We emphasize the distinct integration strategies required by the varying inputs and outputs of short-range geometric GNNs. For detailed insights into the specific implementations within geometric GNNs, we direct readers to the original paper.

## D.1   Pseudocode for Neural P$^3$M Block

The pseudocode for the Neural P$^3$M block is presented in Algorithm 1 as a general framework for iteratively and interdependently updating the atom features $h$ and mesh features $m$. The GNN in the algorithm can incorporate most geometric GNN frameworks that use node features, the atom graph $\mathcal{E}^{\text{short}}$, and edge features $f^{\text{short}}$ as input to update the node features. The FNO serves as the long-range block, updating mesh features according to Eq.17. The representation assignment then calculates the relevant features based on the assignment graph $\mathcal{E}^{\text{assign}}$ and edge features $f^{\text{assign}}$. Finally, the overall representation is updated using information from both the short-range and long-range blocks.

---

**Algorithm 1** Neural P$^3$M block

---

1: **Input:** Atom feature $h^l$, mesh feature $m^l$, atom graph $\mathcal{E}^{\text{short}}$, assignment graph $\mathcal{E}^{\text{assign}}$ and edge features $f^{\text{short}}$, $f^{\text{assign}}$.
2: $\tilde{h}^l \leftarrow \text{GNN}(h^l, \mathcal{E}^{\text{short}}, f^{\text{short}})$ $\qquad\qquad\qquad\qquad$ ▷ Atom2Atom (Short-range)
3: $\tilde{m}^l \leftarrow \text{FNO}(m^l)$ $\qquad\qquad\qquad\qquad\qquad\qquad$ ▷ Mesh2Mesh (Long-range)
4: $(a \leftarrow m)^l_i \leftarrow \text{MLP}\left(\sum_{j \in \mathcal{A}(i)} \tilde{m}^l_j \cdot W^l_{a \leftarrow m} f^{\text{assign}}_{ij}\right)$ $\qquad$ ▷ Mesh2Atom (Repr. Assignment)
5: $(m \leftarrow a)^l_i \leftarrow \text{MLP}\left(\sum_{j \in \mathcal{M}(i)} \tilde{h}^l_j \cdot W^l_{m \leftarrow a} f^{\text{assign}}_{ij}\right)$ $\qquad$ ▷ Atom2Mesh (Repr. Assignment)
6: $h^{l+1} \leftarrow h^l + \tilde{h}^l + \text{LN}((a \leftarrow m)^l)$ $\qquad\qquad\qquad$ ▷ Mesh2Atom (Update)
7: $m^{l+1} \leftarrow m^l + \tilde{m}^l + \text{LN}((m \leftarrow a)^l)$ $\qquad\qquad\qquad$ ▷ Atom2Mesh (Update)
8: **return** $h^{l+1}, m^{l+1}$

---

## D.2   SchNet

SchNet [17] utilized continuous graph convolutional kernels generated from edge features of radial basis functions (RBFs) to capture the geometric information of interatomic distances. In each Neural P$^3$M Block, the atom representations $h^l$ and mesh representations $m^l$ are initially subjected to layer normalization before being processed by a SchNet Block and an FNO Block, respectively.

$$\tilde{h}^l = \text{SchNet Block}(\text{LN}(h^l), ...) \tag{33}$$

$$\tilde{m}^l = \text{FNO}(\text{LN}(m^l)) \tag{34}$$

Following this, the representation assignment block updates these representations separately.

$$(m \leftarrow a)^l = \text{Atom2Mesh}(\tilde{h}^l, ...) \tag{35}$$

$$(a \leftarrow m)^l = \text{Mesh2Atom}(\tilde{m}^l, ...) \tag{36}$$

The exchanged representations are then normalized and combined with their corresponding updated representations via an addition operation. Finally, we employ residual concatenation to obtain the final representation:

$$h^{l+1} = h^l + \tilde{h}^l + \text{LN}((a \leftarrow m)^l) \tag{37}$$

$$m^{l+1} = m^l + \tilde{m}^l + \text{LN}((m \leftarrow a)^l) \tag{38}$$

## D.3   PaiNN

PaiNN [18] is an equivariant graph neural network based on scalar-vector interactions. Each hidden state is described by a tuple of scalar representations $h^l$ and vector representations $\mathbf{vec}^l$ and updated as follows:

$$\tilde{h}^l, \mathbf{\tilde{vec}}^l = \text{PaiNN Block}(\text{LN}(h^l), \mathbf{vec}^l...) \tag{39}$$

We only use scalar representations to exchange information with mesh representations, vector representations can also get long range information when interacting with scalars. The implementations of other parts are consistent with SchNet.

### D.4 DimeNet++

DimeNet++ [9] is an improved version of the original DimeNet [10] architecture. In addition to distance, it further leverages the geometric information of any angles formed by three nodes and applies 2D spherical Bessel functions to embed the angles. Thus, the hidden state $f^l$ of DimeNet++ is at the edge level. To exchange information between atoms and meshes, we need to aggregate the edge-level representations to the node-level representations as follows:

$$\tilde{f}^l = \text{DimeNet Block}(\text{LN}(f^l), ...) \tag{40}$$

$$\tilde{h}_i^l = \sum_{j \in \mathcal{N}(i)} \tilde{f}_{ij}^l \cdot W_{\text{RBF}}^l e^{\text{RBF}}(\|\mathbf{x}_i^a - \mathbf{x}_j^a\|_2) \tag{41}$$

The subsequent implementations are consistent with SchNet, while in order to obtain the final edge-level representations, we combine the atom representations on both sides of the edge, and finally update it as follows:

$$(a_{\text{edge}} \leftarrow m)_{ij}^l = \sigma(W_{\text{concat}} \text{Concat}[(a \leftarrow m)_i^l, (a \leftarrow m)_j^l]) \tag{42}$$

$$f^{l+1} = f^l + \tilde{f}^l + \text{LN}((a_{\text{edge}} \leftarrow m)^l) \tag{43}$$

$$m^{l+1} = m^l + \tilde{m}^l + \text{LN}((m \leftarrow a)^l) \tag{44}$$

### D.5 GemNet

GemNet [8] further extends DimeNet to incorporate geometric information of dihedral angles formed by four atoms and applies high-order Bessel functions to embed the dihedral angles. However, since the computational complexity of quadruplets is too high, GemNet-T used in this paper still uses triplets, which can be viewed as more complex DimeNet. GemNet updates both atom-level and edge-level representations as follows:

$$\tilde{h}^l, \tilde{f}^l = \text{GemNet Block}(\text{LN}(h^l), \text{LN}(f^l), ...) \tag{45}$$

We use node-level representations to exchange information with mesh representations, and subsequent implementations are consistent with SchNet. The final representations are updated as follows:

$$h^{l+1} = \tilde{h}^l \tag{46}$$

$$f^{l+1} = f^l + \tilde{f}^l + \text{LN}((a_{\text{edge}} \leftarrow m)^l) \tag{47}$$

$$m^{l+1} = m^l + \tilde{m}^l + \text{LN}((m \leftarrow a)^l) \tag{48}$$

It should be noted that updates are made solely at the edge-level representations to prevent information redundancy. Our observations indicate that edge-level representations are predominantly parts of GemNet, hence, we focused our updates there. Additionally, we remove the scaling factor from our implementation.

### D.6 ViSNet

ViSNet [23] is an upgraded version of PaiNN, also utilizing scalar-vector interactions that can describe angles, dihedral angles, and improper angles in linear time complexity. When training ViSNet on the MD22 dataset, we find that ViSNet suffers from unstable training when learning rate is relatively large, so we slightly modified the implementation. Unlike the first 4 models, instead of exchanging information using the representations after updating, we use the input representations directly after layer normalization:

$$\tilde{h}^l, \mathbf{v\tilde{e}c}^l, \tilde{f}^l = \text{ViSNet Block}(\text{LN}(h^l), \mathbf{vec}^l, \text{LN}(f^l)) \tag{49}$$

$$\tilde{m}^l = \text{FNO}(\text{LN}(m^l)) \tag{50}$$

$$(m \leftarrow a)^l = \text{Atom2Mesh}(\text{LN}(h^l), ...) \tag{51}$$

$$(a \leftarrow m)^l = \text{Mesh2Atom}(\text{LN}(m^l), ...) \tag{52}$$

The final representations are modified as follows:

$$h^{l+1} = h^l + \tilde{h}^l + (a \leftarrow m)^l \tag{53}$$

$$m^{l+1} = m^l + \tilde{m}^l + (m \leftarrow a)^l \tag{54}$$

$$f^{l+1} = f^l + \tilde{f}^l \tag{55}$$

This modification is similar to altering from post-normalization to pre-normalization in the standard Transformer.

# E   Additional Results

## E.1   Performance of Integrating Equiformer into Neural P$^3$M

We further evaluate the performance of the Equiformer model integrated with our Neural P$^3$M on the two largest molecules on MD22, as shown in Table 5. The results demonstrate our framework's consistent improvement over mainstream state-of-the-art models, further highlighting the compatibility of our approach.

Table 5: Mean absolute errors (MAE) of energy (kcal/mol) and forces (kcal/mol/Å) for the two largest molecules on MD22 compared with Equiformer baseline [14]. The best one in each category is highlighted in **bold**.

| Molecule | Diameter (Å) | | Equiformer | |
|---|---|---|---|---|
| | | | Baseline | Neural P$^3$M |
| Buckyball catcher | 15.89 | energy | 0.3978 | **0.3038** |
| | | forces | 0.1114 | **0.1018** |
| Double-walled nanotube | 32.39 | energy | 1.1945 | **0.6208** |
| | | forces | 0.2747 | **0.2399** |

## E.2   Comparison of Performance on OE62 between LSRM and Neural P$^3$M

Due to the diversity of OE62, the fragmentation algorithm used by LSRM is not suitable for all molecules in this dataset. Nevertheless, for the sake of a thorough comparison, we applied filtering to select a subset of molecules and used this slightly modified dataset to evaluate LSRM's performance on OE62. The results in Table 6 indicate that while LSRM outperforms the baseline, its performance remains below that of our Neural P$^3$M.

Table 6: Energy MAEs on the OE62 dataset compared against the LSRM and baseline models. The best one in each category is highlighted in **bold**.

| Model | Variant | OE62-val | | OE62-test | |
|---|---|---|---|---|---|
| | | MAE meV $\downarrow$ | Rel. % $\uparrow$ | MAE meV $\downarrow$ | Rel. % $\uparrow$ |
| SchNet | Baseline | 133.5 | - | 131.3 | - |
| | LSRM | 72.9 | 45.4 | 72.6 | 44.7 |
| | Neural P$^3$M | **70.2** | **47.4** | **69.1** | **47.4** |
| PaiNN | Baseline | 61.4 | - | 63.3 | - |
| | LSRM | 56.6 | 7.8 | 56.4 | 10.9 |
| | Neural P$^3$M | **54.1** | **11.9** | **52.9** | **16.4** |
| DimeNet++ | Baseline | 51.2 | - | 53.8 | - |
| | LSRM | 47.9 | 6.4 | 50.4 | 6.3 |
| | Neural P$^3$M | **40.9** | **20.1** | **41.5** | **22.9** |
| GemNet-T | Baseline | 51.5 | - | 53.1 | - |
| | LSRM | 50.8 | 1.4 | 51.5 | 3.0 |
| | Neural P$^3$M | **47.2** | **8.3** | **47.4** | **10.7** |

# F    Hyperparameters of Neural P$^3$M

## F.1    Common Hyperparameters

**Ag Dataset**    We use a compact ViSNet which has only a single layer with 128 hidden dimensions and a maximum spherical harmonic order of $l_{\max} = 1$ . For training, we employ the AdamW optimizer with a batch size of 4. The learning rate is dynamically adjusted using the ReduceLROnPlateau scheduler with a decay factor of 0.8, triggered after a patience interval of 30 epochs without improvement. The initial learning rate is set to 0.0018, is preceded by a warm-up phase of 1000 steps. In our loss function, energy and force are weighted at a ratio of 0.1 / 0.9, respectively, to balance their importance during the training process. We employ an early stopping mechanism that terminates training if the validation metric does not improve after 600 epochs. Experiments are conducted on a NVIDIA 16G V100 GPU.

**MD22 Dataset**    We employ a ViSNet that consists of 6 layers with 128 hidden dimensions, and a maximum spherical harmonic order of $l_{\max} = 1$ to enable a fair comparison with LSRM model. We adjust the batch size for each molecule to achieve approximately 1000 steps per epoch (a batch size of 6 for Ac-Ala3-NHMe, 8 for DHA and so on). The initial learning rate is carefully tuned within the range of 0.001 to 0.0018 to optimize performance. Additionally, the weights of energy and force in the loss function is customized for different molecules, with supramolecules using a weight of 0.005 for energy and 0.995 for force, while other molecules using a ratio of 0.05 / 0.95. Other settings remain the same as the Ag dataset. Experiments are conducted on a NVIDIA 16G V100 GPU.

**OE62 Dataset**    Regarding the four models trained on the OE62 dataset, providing a detailed hyperparameters on each is challenging due to their uniqueness. However, to ensure a fair comparison, we set the hyperparameters in line with Ewald MP exactly. The only difference is that after eliminating the scaling factor from the GemNet implementation, we tuned the initial learning rate within the range of 0.0001 to 0.0005. Experiments are conducted on a NVIDIA 80G A100 GPU.

## F.2    Hyperparameters about Mesh Construction

In this subsection, we detail the hyperparameters employed during the mesh construction process. The empirical principles guiding their selection are discussed in Section 4.5, here we focus on the specific hyperparameters in practice.

Table 7: Hyperparameters employed during the mesh construction process on different molecules
.

| Dataset | Molecule | Expand size ($2d$) | Short-range cutoff ($r^{\text{short}}$) | Assignment cutoff ($r^{\text{assign}}$) | $N_x$ | $N_y$ | $N_z$ |
|---|---|---|---|---|---|---|---|
| Ag | - | - | 4.0 Å | 4.0 Å | 3 | 3 | 2 |
| MD22 | Ac-Ala3-NHMe | 1.0 Å | 5.0 Å | 4.0 Å | 3 | 3 | 2 |
| | DHA | 1.0 Å | 5.0 Å | 4.0 Å | 4 | 3 | 2 |
| | Stachyose | 1.0 Å | 4.0 Å | 5.0 Å | 3 | 3 | 2 |
| | AT-AT | 1.0 Å | 5.0 Å | 5.0 Å | 4 | 3 | 2 |
| | AT-AT-CG-CG | 1.0 Å | 5.0 Å | 5.0 Å | 5 | 4 | 3 |
| | Buckyball catcher | 1.0 Å | 4.0 Å | 5.0 Å | 4 | 4 | 2 |
| | Double-walled nanotube | 1.0 Å | 4.0 Å | 5.0 Å | 7 | 3 | 3 |
| OE62 | - | 1.0 Å | 6.0 Å | 4.0 Å | 3 | 3 | 3 |

# G    Profiling Results of Neural P$^3$M on OE62

We present the number of parameters and memory usage (with standard settings and a batch size of 8 of the largest molecule in OE62) as well as the maximum batch size that can be accommodated on a single A100 GPU in the Table 8. The bulk of the memory usage is still attributed to the short-range modules—for instance, 16719 MB versus 19945 MB in GemNet. As anticipated, the integration of the mesh concept and additional modules means that Neural P$^3$M has a higher parameter count and slightly greater memory usage than Ewald MP. Nevertheless, this modest increase in resource demand is offset by the significant performance improvements offered by Neural P$^3$M, along with the computational acceleration brought by FFT.

Table 8: Profiling results on the OE62 dataset compared with Ewald MP and other baseline methods. The best one in each category is highlighted in **bold**.

| Model | Variant | # of Parameters (M) | GPU Memory (MB) | Max. Batch Size |
|---|---|---|---|---|
| SchNet | Baseline | **2.8** | **1623** | **400** |
| | Embeddeds | 14.4 | 1865 | 344 |
| | Cutoff | **2.8** | 1671 | 392 |
| | SchNet-LR | 5.3 | 4835 | 128 |
| | Ewald | 12.2 | 2675 | 240 |
| | Neural P$^3$M | 19.1 | 2283 | 280 |
| PaiNN | Baseline | **12.5** | **8135** | **80** |
| | Embeddeds | 15.7 | 9073 | 72 |
| | Cutoff | **12.5** | 20480 | 32 |
| | SchNet-LR | 15.1 | 11289 | 56 |
| | Ewald | 15.7 | 9901 | 64 |
| | Neural P$^3$M | 28.7 | 11195 | 56 |
| DimeNet++ | Baseline | **2.8** | **12013** | **48** |
| | Embeddeds | 5.4 | 13865 | 40 |
| | Cutoff | **2.8** | 48128 | 8 |
| | SchNet-LR | 3.7 | 13813 | 40 |
| | Ewald | 4.7 | 13725 | 40 |
| | Neural P$^3$M | 6.4 | 17191 | 32 |
| GemNet-T | Baseline | **14.1** | **16719** | **32** |
| | Embeddeds | 16.1 | 17643 | **32** |
| | Cutoff | **14.1** | 33792 | 16 |
| | SchNet-LR | 15.0 | 19131 | **32** |
| | Ewald | 15.8 | 18819 | **32** |
| | Neural P$^3$M | 16.8 | 19945 | **32** |

# H   Ablation Study

We describe our ablation experiments in detail here. We chose the simplest SchNet model and evaluate on the OE62 dataset.

## H.1   Empirical Analysis of the Effectiveness of Atom2Mesh & Mesh2Atom Modules

We set the cutoff distance distance between atoms and mesh points $r^{\text{assign}}$, to a constant value of 4.0 Å and fix the number of discretizations to 3. Subsequently, we incrementally removed either Atom2Mesh, Mesh2Atom, or both from the original architecture to prevent information exchange between short-range and long-range blocks, thereby assessing the impact of these modules. The results are presented in Table 3.

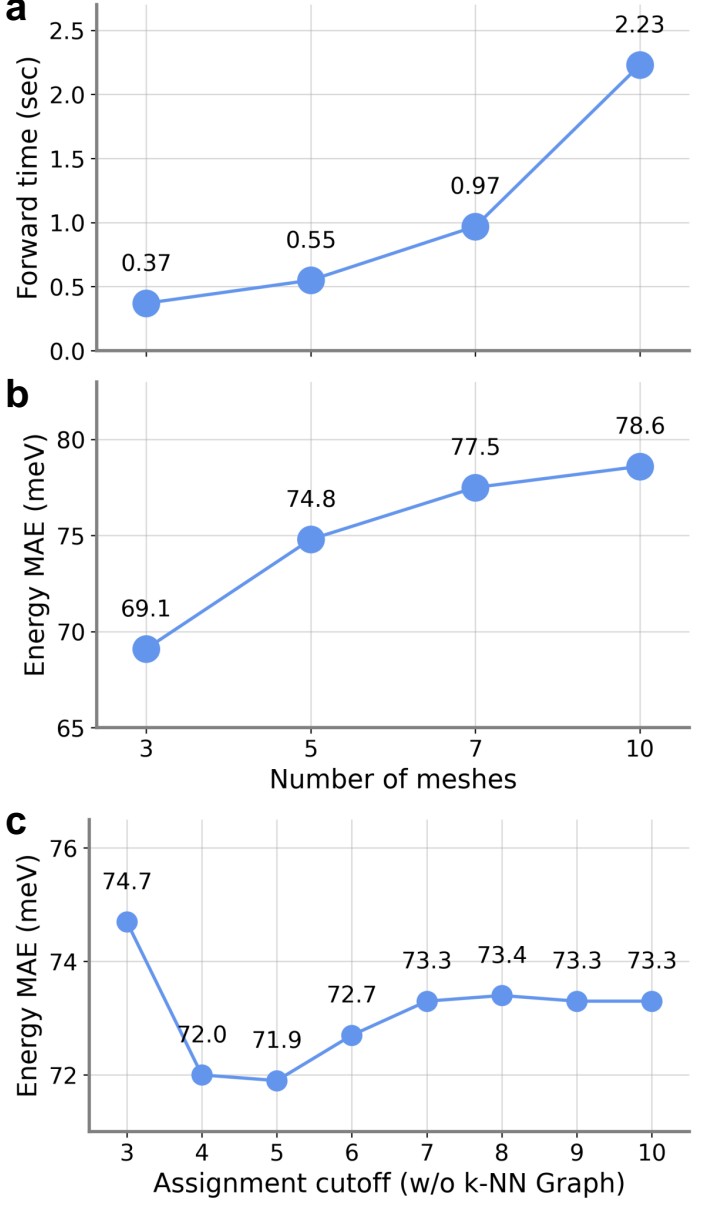

Figure 4: Relationships between the number of meshes and forward time (**a**) and energy MAE (**b**), as well as the relationship between assignment cutoff without k-NN graph and energy MAE (**c**).

## H.2 Empirical Analysis of the Number of Mesh Points

We ensure that the cutoff between the atoms and the mesh points $r^{\text{assign}}$ is constant (4.0 Å) and that the number of discretizations is the same in all three directions, i.e., $N_x = N_y = N_z$. The results of the forward time performance and performance with the number of mesh points are shown in Fig. 4(a) and (b).

## H.3 Empirical Analysis of the Assignment Cutoff Distance

We conduct ablation studies to examine the impact of the assignment cutoff distance. For the performance results reported in Table 2 , we use a combination of a radius graph and k-NN graph, setting the maximum number of neighbors to 5, which generally minimizes multiple assignments. To assess the effect of multiple assignments, this ablation experiment uses only the radius graph, varying the assignment cutoff distance from 3 to 10. The performance results based on different assignment cutoff distances are shown in Fig. 4(c).

