# OpenReview forum: "Neural P$^3$M: A Long-Range Interaction Modeling Enhancer for Geometric GNNs"
_NeurIPS.cc/2024/Conference — NeurIPS 2024 poster_

### Official Review · Reviewer_g8KM · 2024-07-09

**Soundness:** 3
**Presentation:** 3
**Contribution:** 3
**Rating:** 7
**Confidence:** 3

**Summary:**

The accurate modeling of both short-range and long-range interactions in molecular systems is crucial for predicting properties like molecular energies and forces with high precision. However, traditional Geometric Graph Neural Networks (GNNs) fail to capture such interaction. The paper introduces Neural $\text{P}^3$M, a framework that enhances the capabilities of GNNs via meshing up the Ewald summation. Compared to previous Ewald-based GNNs, the Neural$\text{P}^3$M further enhances the model by incorporating mesh and atom representation in a trainable manner. The paper conducts extensive empirical studies and justifies the effectiveness of the proposed method.

**Strengths:**

- Unlike the Ewald method, which primarily handles electrostatic interactions, Neural P3M is designed to enhance the modeling of both short-range and long-range interactions in molecular systems.
- The Atom-Mesh interaction mechanism employed by Neural $\text{P}^3$ M, along with the Fast Fourier Transform (FFT), is generally faster than the traditional Ewald method, especially for larger systems. The efficiency of such a design is demonstrated by the comparison of the running time in Table 2.
- The paper is generally well-written, with clear illustrations and tables.

**Weaknesses:**

- The major contribution of the Neural $\text{P}^3$ M seems to lie in the representation assignment. However, the paper does provide enough ablation studies on the effectiveness of this component. It is unclear whether changing the Fourier Transform to FFT is the main reason for the efficiency.
- The introduction of different concepts can be further improved. For example, the detailed introduction of the Ewald summation in Section 2 mostly is not explicitly ref in the following section. However, Equation 17, which is crucial for the Neural $\text{P}^3$ M, is not elaborated properly, especially for the influence function G and the connection to Equation 9.
- Despite the parameterized assigning function, the representation assignment still assigns each atom to multiple mesh points when the number of mesh points increases. This would post drawbacks on the performance and forward time. The author could consider discussing the different choices of the distance cut-off, as this hyperparameter could differ across datasets.

**Questions:**

1. Discuss the different choices for the distance cut-off.
2. Could the author explain more about the connection between Equation 17 and Equation 9 and the concept of the parameterizing strategy of influence function G?
3. Provide ablation studies on the proposed component and the discussion on the effectiveness of the FFT.

**Limitations:**

It is interesting to involve mesh points to help capture the long-range interaction. The author could extend the idea to other molecules like protein or DNA.

---

> ### Author Rebuttal · Authors · 2024-08-06
>
> We thank you for your recognition of our work’s contribution and clear organization. We will address your questions and concerns as follows.
>
> **Weakness 1, Ablation Study of Atom2Mesh & Mesh2Atom and Efficiency of FFT**
>
> We provide additional ablation studies on the impact of Atom2Mesh & Mesh2Atom modules in the following table under settings for training variants of SchNet on OE62 dataset. It can be shown that both modules indeed contribute to the final performance in a synergic fashion (Introducing just a single module results in poorer results).
>
> | Architecture | Test EMAE |
> | --- | --- |
> | Without Both | 72.07 |
> | Without Mesh2Atom | 76.14 |
> | Without Atom2Mesh | 74.48 |
> | Neural P$^3$M | **69.10** |
>
> Regarding efficiency, we would like to highlight that our approach, which incorporates mesh points, draws **inspiration** from the FFT applied to Ewald summation. In practice, we employ the FNO as the model, which allows for efficient computation. While the inclusion of mesh nodes does introduce a larger number of parameters, this is offset by the significant reduction in runtime compared to Ewald MP. This improvement is largely due to the efficiency afforded by the FFT. For a more detailed analysis, please refer to our profiling results presented in Table 3 in the Global Rebuttal and Table 2 in our manuscript.
>
> **Weakness 2, More Explanation of the Parameterizing Long-Range Interaction**
>
> Sorry for the lack of clarity. We elaborate more on our choice of using FNO for capturing the long-range interactions. Eq.9 indicates that the energy of long-range interactions can be regarded as the result of the convolution between the influence function $G$ and the charge density $\rho_M$. According to the convolution theorem, we can accelerate the convolution in the Fourier domain by point-wise multiplication. If we consider the charge density $\rho_m$ as a representation of mesh $m^l$ and parameterize $G$ directly in the Fourier domain $\tilde{G}$, we can obtain the latter part of Eq.17. As for the first part, it is merely a gated connection.
>
> **Weakness 3, Ablation Study of Assignment Cutoff**
>
> We provide additional ablation studies on the impact of the assignment cutoff distance as follows under settings for training SchNet on OE62 dataset. For the performance (69.1) reported in the paper, we use a combination of radius graph and k-NN graph, setting the maximum number of neighbors to 5, which generally results in minimal multiple assignments. To test the effect of multiple assignments, we use only the radius graph in this ablation experiment, and we observe that all the experiments perform slightly worse due to multiple assignments. However, an appropriately chosen cutoff (4 or 5 Å) still yields relatively optimal results. Notably, the results do not deteriorate further as the assignment cutoff increases. We hypothesize that this may be because a larger assignment cutoff creates a larger neighborhood environment, making it easier to learn the assignment function with the fixed number of meshes, thereby mitigating the multiple assignments problem.
>
> |  Assignment cutoff (Without k-NN Graph) | Test EMAE |
> | --- | --- |
> | 3 | 74.7 |
> | 4 | 72.0 |
> | 5 | 71.9 |
> | 6 | 72.7 |
> | 7 | 73.3 |
> | 8 | 73.4 |
> | 9 | 73.3 |
> | 10 | 73.3 |
>
> **Questions 1**
>
> See Weakness 3 for details.
>
> **Questions 2**
>
> See Weakness 2 for details.
>
> **Questions 3**
>
> See Weakness 1 for details.
>
> **Limitation 1, Extending to DNA/Protein**
>
> Thanks for your insightful suggestions. We acknowledge that we mainly focus on testing the effectiveness of our Neural P$^3$M framework in this work so we primarily experimented with molecules with abundant datasets and baselines. We noted that, currently, energy/force datasets for large biomolecules are scarce, as they often require expensive ab initio calculation. We will leave this scaling up as one of the future directions of this work.

---

> > ### Comment · Reviewer_g8KM · 2024-08-09
> >
> > Thanks for your detailed response. It addresses all my problems!

---

> > > ### Author Response · Authors · 2024-08-09
> > > **Gratitude for Your Review**
> > >
> > > We really appreciate your recognition of our work and your decision. Thank you for the effort and time you put into your review!

---

### Official Review · Reviewer_iSFT · 2024-07-11

**Soundness:** 3
**Presentation:** 3
**Contribution:** 3
**Rating:** 6
**Confidence:** 4

**Summary:**

This work introduces a long-range focused GNN that utilizes the combination of atom and mesh representations. The mesh framework in this work is trainable and unconstrained to the fragmentation algorithm. Results demonstrate superior performances across MD22, Ag, and OE62 datasets.

**Strengths:**

- The need for long-range GNN methods is crucial for various molecular representation tasks
    - This work tackles this pertinent issue utilizing a novel combination of atom and mesh representation
- The underlying method is well-described
- The distinction of this approach to other recent and relevant approaches is well-described
- The paper is well-written and easy to walk through
- Strong results
    - Demonstrates improvement over vanilla VisNet on Ag dataset
    - SOTA results on most targets and molecules of MD22 and gives significant improvements on larger molecules
    - On OE62 dataset, NeuralP3M performs better on all architectures over other approaches. It’s also faster than Ewald (for most cases) due to FFT which is impressive.

**Weaknesses:**

- Few of the results seem carefully designed to demonstrate improvements through this approach which makes the overall impact unclear. I've mentioned more concrete points related to this as questions and I can be convinced otherwise post rebuttal/discussion.
- The anonymous link to the code is not in the paper.

**Questions:**

- What’s the reason for choosing VisNet to add Neural P3M in Table 1 instead of Equiformer/MACE which has better baselines?
- MD22 results seem to have very carefully designed hyperparameters (cutoffs and number of meshes in each direction) in Table 4 of Appendix E.2. Do you have thoughts on the feasibility of finding these hyperparams on datasets with larger diversity? Additionally, what do the results look like if you just have a uniform hyperparameter across all molecules?
- Is it possible to add results with LSRM in Table 2 results as well? Was there a reason for not including that?
- In Table 2, how does the GPU memory usage compare across these methods? Just out of curiosity, I wanted to know what were the max. batch sizes you were able to fit across these methods. That would most likely make throughput comparison with Ewald even more impressive (assuming a smaller batch size fit for Neural P3M)
- How important is it to have both atom2mesh and mesh2atom modules? Do you get similar performances by just having one and maybe some speedup? Some of the prior literature in charge density models only have mesh2atoms message passing.

**Limitations:**

Yes, this work does mention its limitations and potential negative impact.

---

> ### Author Rebuttal · Authors · 2024-08-06
>
> We thank your recognition of our work’s novelty, contribution and clear organization. We will address your questions and concerns as follows.
>
> **Weakness 1, Evaluation & Improvement**
>
> We resolve the issues point-to-point in the **Questions** Section.
>
> **Weakness 2, Anonymized Code**
>
> We intend to make our code publicly available upon acceptance of our manuscript. For the purpose of this review process, we have provided an anonymized version of the code https://anonymous.4open.science/r/Neural_P3M-1552.
>
> **Question 1, ViSNet as the Baseline**
>
> We selected ViSNet as our base model for MD22 because LSRM demonstrated better performance using ViSNet than Equiformer across molecules. This allows us to make a direct and equitable comparison between our results on ViSNet and those reported by LSRM. Additionally, we present further results for Equiformer on MD22 dataset. The incorporation of Neural P$^3$M has enhanced the performance of Equiformer, even surpassing that of ViSNet (shown in the following table). Given the constraints of resources, we are presenting results only for the two largest molecules on MD22 at this moment.  We will provide a complete evaluation if our work gets accepted.
>
> | Molecule |  | Baseline | Neural P$^3$M |
> | --- | --- | --- | --- |
> | Buckyball catcher | Energy | 0.3978 | **0.3038** |
> |  | Forces | 0.1114 | **0.1018** |
> | Double-walled nanotube | Energy |  1.1945 | **0.6208** |
> |  | Forces | 0.2747 | **0.2399** |
>
> **Question 2, Choice of Mesh & Cutoff**
>
> As the molecules in MD22 range from peptides to nanotubes with diverse molecule sizes, each individual molecule is a separate dataset. We chose the number of mesh points and the cutoff distance accordingly in a **pre-defined and consistent manner**. Actually, we didn't tune these hyperparameters due to the vastness of the search space. In practice, we typically set the assignment cutoff distance at 4.0 or 5.0 Å, ensuring that the product of the number of mesh points and the cutoff is **roughly equivalent to the cell size in each dimension**. For example, the tubular nanotube would need more mesh points along its longest dimension $x$. For a dataset with larger diversity containing different molecules, OE62 is such an example, for which we have demonstrated the performance improvement of Neural P$^3$M using the average cell size following the same manner.
>
> **Question 3, LSRM on OE62**
>
> We initially did not include this result because of the diversity of OE62, as the fragmentation algorithm employed in LSRM isn't suitable for all molecules in OE62. Nonetheless, by filtering some molecules (validation:285, test:293, train:2387) and utilizing a marginally different dataset, we have also presented the performance of LSRM on OE62 dataset (See Table 1 in the Global Rebuttal). These additional results suggest that LSRM surpasses the baseline, yet it does not perform as well as our Neural P$^3$M.
>
> **Question 4, GPU Memory Usage**
>
> See Table 3 and its discussion in the Global Rebuttal for more details.
>
> **Question 5, Ablation Study of Atom2Mesh & Mesh2Atom**
>
> We provide additional ablation studies on the impact of Atom2Mesh & Mesh2Atom modules in the following table under settings for training variants of SchNet on OE62 dataset. It can be shown that both modules indeed contribute to the final performance in a synergic fashion (Introducing just a single module results in poorer results). The cost of two modules is very little and the bottleneck is in the short-range model, so there is no significant speedup.
>
> | Architecture | Test EMAE |
> | --- | --- |
> | Without Both | 72.07 |
> | Without Mesh2Atom | 76.14 |
> | Without Atom2Mesh | 74.48 |
> | Neural P$^3$M | **69.10** |

---

> > ### Comment · Reviewer_iSFT · 2024-08-09
> > **Response**
> >
> > I thank the authors for their rebuttal. Overall, all my raised concerns have been addressed. As a result, I've increased my score.

---

> > > ### Author Response · Authors · 2024-08-09
> > > **Gratitude for Your Review and Increasing the score**
> > >
> > > Thank you for taking the time. We really appreciate your decision. Your support is invaluable in improving our work.

---

### Official Review · Reviewer_7zWV · 2024-07-13

**Soundness:** 3
**Presentation:** 2
**Contribution:** 2
**Rating:** 6
**Confidence:** 4

**Summary:**

The paper introduces Neural P3M, a framework designed to enhance geometric GNNs by incorporating mesh points alongside atoms and transforming traditional mathematical operations into trainable components. The mesh representations offers discrete resolutions necessary for formulating long-range terms. The Neural P3M is also efficient due to the reduced computational complexity afforded by FFT.
The paper starts with highlighting the importance of long-range terms, which are absent and inefficient in previous works. The paper then explain the preliminary of Ewald summation and the meshing up methods with detailed formulas. The novel methods and neural network blocks are presented in the next section and evaluated with various models and datasets.
The experiment results show significant improvements. When integrated with ViSNet, Neural P3M achieves state-of-the-art performance in energy and force predictions across several large molecules, outperforming other leading models. The framework, combined with models like SchNet, PaiNN, DimeNet++, and GemNet-T, demonstrates enhanced performance and faster computation times compared to related works.
Neural P3M provides a robust framework for enhancing geometric GNNs, enabling them to capture long-range interactions efficiently. The framework's adaptability to various molecular systems and its demonstrated performance improvements on key benchmarks make it a significant contribution to the field. The study also highlights areas for future research, such as optimizing the number of mesh points and exploring alternatives to FFT for modeling long-range interactions.

**Strengths:**

1. Neural P3M effectively integrates mesh points alongside atoms, which allows it to capture long-range interactions more accurately than traditional GNNs. This enhancement addresses a significant limitation in current molecular modeling approaches, particularly for large molecular systems.
2. The framework is built upon well-established principles such as Ewald summation and P3M methods. This theoretical grounding lends credibility to the approach.
3. Neural P3M is designed to be a versatile enhancer that can be integrated with a wide range of existing geometric GNN architectures, including SchNet, PaiNN, DimeNet++, and GemNet-T. This compatibility ensures that the framework can be widely adopted and used in different contexts.
4. Neural P3M reduces the computational complexity of long-range interaction calculations, making it feasible to handle large-scale systems efficiently. The framework also exhibits a faster computation time than Ewald MP.
5. Its theoretical soundness, empirical success, and detailed implementation make it a valuable contribution to the field of molecular modeling.

**Weaknesses:**

1. The framework's reliance on complex mathematical operations and integration of mesh-based methods with GNNs can make implementation challenging. Researchers and practitioners may require significant expertise in both GNNs and numerical methods to effectively utilize Neural P3M. It is better to remove some unnecessary equations in Section 2 and 3, or move them to the appendix or references.
2. Additionally, you can present the P3M Blocks by some pseudocode.
3. The framework's need to handle both atomic and mesh representations simultaneously may lead to increased memory usage, which could be a bottleneck for handling large datasets or systems with limited hardware capabilities. You can present the GPU memory usage while training model with or without Neural P3M.
4. There are some trivial mistakes in Table 2. If the higher Rel. is better, the up arrows should be used. The best runtime should also be highlighted.

**Questions:**

1. Does Neural P3M have some restrictions of geometric GNN models? Or it can be combined with most GNNs?
2. What are the throughputs of Short-Range Block and Long-Range Block? Is the Long-Range Block slower than Shaor-Range Block? Which block is the main bottleneck or is it possible to improve the performance?
3. The variants of the same model usually share some common layers or blocks. Is it possible to reuse or frozen some common layers in a pre-trainded model, and fine-tune the newly added blocks, such as Long-Range Block? So the training performance will be further improved.

**Limitations:**

1. Distributed training is important and efficient while training the general models with large datasets. However, the distributed training of Neural P3M is not evaluated, and the proposed models and results are now limited to one GPU.
2. Profiler results and the number of parameters in each variant are not presented.

---

> ### Author Rebuttal · Authors · 2024-08-06
>
> We thank you for your recognition of our work’s novelty and contribution to the field of molecular modeling. We will address your questions and concerns as follows.
>
> **Weakness 1, Mathematical Details**
>
> Thank you for your suggestions. We have relocated some non-essential content from Sections 2 and 3 to the appendix to better organize the paper. Additionally, we intend to release the code upon acceptance of our work to make implementation easier for all researchers.
>
> **Weakness 2, Pseudocode**
>
> Thank you for your suggestions. We have included pseudo-code in the appendix to further illustrate our method. However, due to the constraints of OpenReview, it is not feasible to display it directly. For your convenience, we have also provided anonymized reference code (https://anonymous.4open.science/r/Neural_P3M-1552) to facilitate better understanding of our work.
>
> **Weakness 3, Computational Complexity**
>
> Thank you for your suggestions. We detail the memory usage in Table 3 in the Global Rebuttal. The bulk of the memory usage is still attributed to the short-range modules—for instance, 16719 MB versus 19945 MB in GemNet. This level of memory consumption is considered acceptable in light of the performance gains achieved.
>
> **Weakness 4, Typos in Table 4**
>
> Thanks for pointing these out. We have fixed them in the revised manuscript.
>
> **Question 1, Restriction on GNN**
>
> We have integrated Neural P$^3$M with a wide range of geometric GNNs in our experimental settings. This included both classic GNNs like SchNet and DimeNet and more recent and advanced equivariant architectures like PaiNN, GemNet, Equiformer, and ViSNet. In this way, our framework is a general one that can be integrated into most geometric GNNs.
>
> **Question 2, Computational Bottleneck**
>
> We have provided the empirical evaluation of the runtime in the following table. The runtime for the long-range interactions remains consistent, whereas the short-range interactions are the primary bottleneck, particularly as the model for these interactions grows in complexity. Efforts to reduce computational demands, such as implementing the density trick to reduce the cost of many-body interactions, have been investigated in models like ViSNet [1] and MACE [2]. Additionally, simplifying the complexity of the CG-product is another avenue being explored to speed up short-range interaction models [3].
>
> | Base Model | Short Range Forward Time | Long Range Forward Time |
> | --- | --- | --- |
> | SchNet | 0.4797ms | 0.9814ms |
> | PaiNN | 0.8986ms | 0.9827ms |
> | DimeNet++ | 0.9761ms | 0.9702ms |
> | GemNet-T | 2.9914ms | 0.9817ms |
>
> [1]  Y, Wang T, Li S, et al. Enhancing geometric representations for molecules with equivariant vector-scalar interactive message passing[J]. Nature Communications, 2024, 15(1): 313.
>
> [2] Batatia I, Kovacs D P, Simm G, et al. MACE: Higher order equivariant message passing neural networks for fast and accurate force fields[J]. Advances in Neural Information Processing Systems, 2022, 35: 11423-11436.
>
> [3] Passaro S, Zitnick C L. Reducing SO (3) convolutions to SO (2) for efficient equivariant GNNs[C]//International Conference on Machine Learning. PMLR, 2023: 27420-27438.
>
> **Question 3, Freezing Common Layers**
>
> Following your recommendations, we evaluated the performance of our base GNN when it is first pre-trained and subsequently fine-tuned under settings for training variants of SchNet on OE62 dataset. The results are presented in the following table. We observed that freezing the pre-trained weights results in a notable decline in performance. Conversely, permitting fine-tuning of the weights yields a marginal improvement. This could be attributed to the possibility that the energy is an amalgamation of long-range and short-range interactions, suggesting that the short-range representations, when trained directly with energy labels, may not be ideally suited for direct integration with long-range components. Utilizing pre-trained short-range models is an intriguing area for future research and is something we plan to explore further.
>
> |  | Test EMAE |
> | --- | --- |
> | Freeze | 83.9 |
> | Finetuned | **68.6** |
> | From scratch | 69.1 |
>
> **Limitations 1, Distributed Training**
>
> Our framework is built on top of PyTorch Lightning, which allows for straightforward extension to Distributed Data Parallel (DDP) mode. We have conducted experiments under the setting for training SchNet + Neural P$^3$M on the large OC20-2M dataset with 4 GPUs, and our results demonstrate significant improvements, underlining the robustness of our framework.
>
> |  | Baseline | Cutoff | SchNet-LR | Ewald | Neural P$^3$M |
> | --- | --- | --- | --- | --- | --- |
> | EMAE | 895 | 869 | 984 | 830 | **693** |
> | FMAE | 61.1 | 60.3 | 65.3 | 56.7 | **55.6** |
>
> **Limitations 2, Profiler results**
>
> See Table 3 and its discussion in the Global Rebuttal for more details.

---

> > ### Author Response · Authors · 2024-08-12
> > **We are looking forward to your reply**
> >
> > Dear Reviewer 7zWV,
> >
> > We are thankful for your valuable feedback and the recognition you have given our manuscript.
> >
> > **As the deadline for discussion nears**, we would like to gently remind you of the thorough response we have crafted to address the issues you highlighted. We have carefully addressed **all the points you raised in the Weaknesses**, and have conducted **additional experiments** to answer your interesting questions. We find your questions heuristic and are happy to further engagement on these topics.
> >
> > If you believe we have resolved the issues and adequately answered your questions, we would be grateful if you could reconsider your score. We welcome any further questions or discussions you may wish to have.
> >
> > Warm regards,
> >
> > The Authors

---

> > > ### Comment · Reviewer_7zWV · 2024-08-14
> > > **Reply to Rebuttal**
> > >
> > > Thank you for your answers to my questions, suggestions, and weaknesses. Your answers have clarified my doubts to a considerable extent. I will not increase the rate of the paper.

---

> > > > ### Author Response · Authors · 2024-08-14
> > > > **Gratitude for Your Review**
> > > >
> > > > Thank you once again for your recognition of our work and for the time and effort you've dedicated to the review process. We are very happy to receive your feedback and are pleased to know that we've addressed your concerns. Your insightful questions and suggestions are intriguing and will certainly inspire our future work.

---

### Official Review · Reviewer_M1Fm · 2024-07-16

**Soundness:** 2
**Presentation:** 3
**Contribution:** 2
**Rating:** 3
**Confidence:** 3

**Summary:**

This work proposes Neural P3M, a framework that enhances geometric GNNs by integrating mesh points and leveraging Fast Fourier Transform (FFT) for efficient computation of long-range interactions. The framework includes short-range and long-range interaction modeling and enables the exchange of information between atom and mesh scales. Neural P3M improves the prediction of energies and forces in large molecular systems, achieving good performance on benchmarks like MD22 and OE62.

**Strengths:**

The proposed framework is capable of being incorporated in short-range geometric GNNs, although distinct integration strategies are needed due to the varying inputs and outputs of different models.

The improvements in benchmarks are promising, which demonstrates the power of the proposed model.

The paper is well-structured and clearly written, making it accessible to readers without knowledge of the related concepts like Ewald summation.

**Weaknesses:**

1. I feel the overall impact and novelty of this work are limited, given that several works have adopted Ewald summation in geometric GNNs. This work enhanced this concept by introducing FFT for accelerated Ewald summation, which is a common way in traditional simulations and is also already mentioned as a possible direction in [1].

2. The experimental part is not comprehensive. The experiment on MD22 doesn't have a comparison to [1], and LSRM is not being compared on OE62. Besides, it is important to also compare the memory consumption between different approaches when dealing with long-range interactions, since in many real-world problems we hope to capture long-range interactions in large molecular systems.

[1] Ewald-based long-range message passing for molecular graphs. ICML 2023

**Questions:**

1. What do the "Embeddings", "Cutoff", and "SchNet-LR" mean in Table 2?

2. What is the computational complexity of the proposed method?

**Limitations:**

The authors have addressed the limitations and potential societal impacts.

---

> ### Author Rebuttal · Authors · 2024-08-06
>
> We thank you for your recognition of our work’s superior performance and clear organization. We will address your questions and concerns as follows.
>
> **Weakness 1, Novelty & Contribution of Neural P$^3$M**
>
> While it's true that FFT is commonly employed in traditional chemical computations, as discussed in Section 2. However, incorporating such a technique as a learnable component within a Geometric GNN framework is **far from straightforward**. To the best of our knowledge, we introduce the **novel mesh concepts** for energy and force prediction, marking one of our major contributions. In terms of model architecture, we have also introduced the **innovative learnable Atom2Mesh and Mesh2Atom modules**, which are designed to enhance the exchange of information between short-range atomic and long-range mesh representations. With the integration of mesh nodes, FFT emerges as a natural choice for modeling long-range interactions. However, we could also opt for alternative networks such as transformers or 3D CNNs. All of these carefully designed components set Neural P$^3$M apart from existing Ewald-based models like Ewald MP. Thus, our framework not only expands upon Ewald-based approach but also paves the way for a new class of mesh-based methodologies in the modeling of long-range interactions in 3D molecular structures.
>
> **Weakness 2, Additional Experimental Results**
>
> Thanks you for your suggestions. We've implemented LSRM in our experiments on OE62 dataset as well as in Ewald MP applied to ViSNet on MD22 dataset (across a total of **11** experiments). Details can be found in Tables 1 and 2 in the Global Rebuttal. While both Ewald MP and LSRM outperform the baseline, Neural P$^3$M consistently delivers superior performance in the majority of the test baseline models (OE62) and molecules (MD22).
>
> For the memory consumption, please see Table 3 and its discussion in the Global Rebuttal for more details.
>
> **Question 1, Concepts in Table 2**
>
> The experimental results of different variants of the base models came from the original Ewald MP [1] paper. “Embedding” indicates a model with a larger embedding dimension, “Cutoff” for a larger cutoff distance, and “SchNet-LR” for models with the pairwise LR block. These variants are existing improvement methods for the base model.
>
> [1] Ewald-based long-range message passing for molecular graphs. ICML 2023
>
> **Question 2, Computational Complexity**
>
> The computational complexity for the long-range interactions scales with $O(K\log K D^2)$, where  $K$ represents the number of mesh points and $D$ is the dimensionality of the hidden layers. The complexity of the short-range interactions is contingent upon the chosen GNN architecture, typically scaling with $O(|\mathcal{E}| D^2)$, where $\mathcal{E}$ denotes the set of edges. The overhead introduced by the Atom2Mesh and Mesh2Atom modules is negligible when compared to the computational demands of the long-range and short-range components. Notably, the computational bottleneck is primarily determined by the selected number of mesh points and the volume of atom edges. In practical applications, we observed that setting $K$ to $3^3=27$ yielded a significant enhancement in performance relative to the baseline model.

---

> > ### Author Response · Authors · 2024-08-12
> > **We are looking forward to your reply**
> >
> > Dear Reviewer M1Fm,
> >
> > Thank you for your insightful feedback on our manuscript.
> >
> > **As the deadline for discussion nears**, we wish to remind you that we have provided **a comprehensive response** to address the concerns you raised. With respect to the novelty of our work, we would like to highlight that the other reviewers have recognized our contributions and novelty. It seems there may have been a **misunderstanding** regarding this during the initial review, and we highlighted our novelty in the previous rebuttal. Furthermore, in response to your comments on our experiments, we have made the necessary enhancements.
> >
> > If you believe we have resolved the issues, we would be grateful if you could reconsider your score. We welcome any further questions or discussions you may wish to have.
> >
> > Warm regards,
> >
> > The Authors

---

### Author Rebuttal · Authors · 2024-08-06

We would like to express our sincere gratitude to all the reviewers for dedicating their time to read our manuscript and for offering their valuable suggestions. We appreciate the recognition our manuscript has received from the reviewers. We have also addressed each concern raised on a point-by-point response through a total of 28 experiments and a comprehensive profiling. Here we show some common results for all reviewers.

**Results 1, Performance of LSRM on OE62 Dataset**

Due to the LSRM's limited generalizability and its dependency on the fragmentation algorithm, we have excluded certain molecules from OE62 dataset (validation:285, test:293, train:2387). We present the performance results (in the format of *Valid EMAE/Test EMAE*) of the LSRM on this slightly modified dataset.

|  | Baseline | LSRM | Neural P$^3$M |
| --- | --- | --- | --- |
| SchNet | 133.5/131.3 | 72.9/72.6 | **70.2/69.1** |
| PaiNN | 61.4/63.3 | 56.6/56.4 | **54.1/52.9** |
| DimeNet++ | 51.2/53.8 | 47.9/50.4 | **40.9/41.5** |
| GemNet-T | 51.5/53.1 | 50.8/51.5 | **47.2/47.5** |

**Results 2, Performance of Ewald MP on MD22 Dataset**

We present the performance results (*MAE*) of Ewald MP combined with ViSNet on MD22 dataset.

| Molecule |  | Baseline | Ewald | Neural P$^3$M |
| --- | --- | --- | --- | --- |
| Ac-Ala3-NHMe | Energy | 0.0796 | 0.0775 | **0.0719** |
|  | Forces | 0.0972 | 0.0814 | **0.0788** |
| DHA | Energy | 0.1526 | 0.0932 | **0.0712** |
|  | Forces | 0.0668 | **0.0664** | 0.0679 |
| Stachyose | Energy | 0.1283 | 0.1089 | **0.0856** |
|  | Forces | **0.0869** | 0.0976 | 0.0940 |
| AT-AT | Energy | 0.1688 | 0.1487 | **0.0714** |
|  | Forces | 0.1070 | 0.0885 | **0.0740** |
| AT-AT-CG-CG | Energy | 0.1995 | 0.1571 | **0.1124** |
|  | Forces | 0.1563 | 0.1115 | **0.0993** |
| Buckyball catcher | Energy | 0.4421 | 0.3575 | **0.3543** |
|  | Forces | 0.1335 | 0.0989 | **0.0846** |
| Double-walled nanotube | Energy | 1.0339 | 0.7909 | **0.7751** |
|  | Forces | 0.3959 | 0.2875 | **0.2561** |

**Results 3, Profiling results of Neural P$^3$M**

We present the number of parameters and memory usage (with standard settings and a batch size of 8 of the largest molecule in OE62) as well as the maximum batch size that can be accommodated on a single A100 GPU in the following table. The bulk of the memory usage is still attributed to the short-range modules—for instance, 16719 MB versus 19945 MB in GemNet. As anticipated, the integration of the mesh concept and additional modules means that Neural P$^3$M has a higher parameter count and slightly greater memory usage than Ewald MP. Nevertheless, this modest increase in resource demand is offset by the significant performance improvements offered by Neural P$^3$M, along with the computational acceleration brought by FFT.

|  |  | # of Parameters | GPU Memory (MB) | Max Batch Size |
| --- | --- | --- | --- | --- |
| SchNet | Baseline | 2.8M | 1623 | 400 |
|  | Embeddings | 14.4M | 1865 | 344 |
|  | Cutoff | 2.8M | 1671 | 392 |
|  | SchNet-LR | 5.3M | 4835 | 128 |
|  | Ewald | 12.2M | 2675 | 240 |
|  | Neural P$^3$M  | 19.1M | 2283 | 280 |
| PaiNN | Baseline | 12.5M | 8135 | 80 |
|  | Embeddings | 15.7M | 9073 | 72 |
|  | Cutoff | 12.5M | 20480 | 32 |
|  | SchNet-LR | 15.1M | 11289 | 56 |
|  | Ewald | 15.7M | 9901 | 64 |
|  | Neural P$^3$M  | 28.7M | 11195 | 56 |
| DimeNet++ | Baseline | 2.8M | 12013 | 48 |
|  | Embeddings | 5.4M | 13865 | 40 |
|  | Cutoff | 2.8M | 48128 | 8 |
|  | SchNet-LR | 3.7M | 13813 | 40 |
|  | Ewald | 4.7M | 13725 | 40 |
|  | Neural P$^3$M | 6.4M | 17191 | 32 |
| GemNet-T | Baseline | 14.1M | 16719 | 32 |
|  | Embeddings | 16.1M | 17643 | 32 |
|  | Cutoff | 14.1M | 33792 | 16 |
|  | SchNet-LR | 15.0M | 19131 | 32 |
|  | Ewald | 15.8M | 18819 | 32 |
|  | Neural P$^3$M | 16.8M | 19945 | 32 |

---

### Author Response · Authors · 2024-08-14
**Conclusion for Author-Reviewer Discussion Period**

Dear Reviewers,

We sincerely thank you for your insightful and constructive feedback throughout the review process. Your thoughtful comments have helped make our work more comprehensive and concrete. We are grateful for your recognition of our efforts, contributions, and novelty in our Neural P$^3$M framework, and we are glad that our rebuttals with additional 28 experiments and profiling results have successfully addressed your questions and comments.

We especially enjoyed the interactive discussions during the rebuttal period and we always look forward to the possibility of engaging in a productive dialogue with the reviewers. We are grateful to three reviewers for their active engagement in the discussion period and for their unanimous recognition of our contributions.

We will make sure to add the additional experiment results and ablation studies in our revised manuscript. Again, we thank you for your valuable input and for helping us bring this research to fruition.

Warm Regards,

The Authors

---

### Decision · Program_Chairs · 2024-09-25

**Decision:**

Accept (poster)

**Comment:**

This paper consider the incorporation of mesh points into geometric GNNs for materials science. The reviewers largely agree that this is a valuable contribution that is well-supported, and I believe concerns have been adequately addressed. I accordingly recommend acceptance.